# Expression and Immune Response Profiles in Nile Tilapia (*Oreochromis niloticus*) and European Sea Bass (*Dicentrarchus labrax*) During Pathogen Challenge and Infection

**DOI:** 10.3390/ijms252312829

**Published:** 2024-11-28

**Authors:** Ahmed A. Saleh, Asmaa Z. Mohamed, Shaaban S. Elnesr, Asmaa F. Khafaga, Hamada Elwan, Mohamed F. Abdel-Aziz, Asmaa A. Khaled, Elsayed E. Hafez

**Affiliations:** 1Animal and Fish Production Department, Faculty of Agriculture (*Al-Shatby*), Alexandria University, Alexandria 11865, Egypt; 2Animal and Fish Production Department, Faculty of Agriculture (*Saba Basha*), Alexandria University, Alexandria 21531, Egypt; asmaazayed@alexu.edu.eg (A.Z.M.); dr.asmaa_khaled@alexu.edu.eg (A.A.K.); 3Department of Poultry Production, Faculty of Agriculture, Fayoum University, Fayoum 63514, Egypt; ssn00@fayoum.edu.eg; 4Department of Pathology, Faculty of Veterinary Medicine, Alexandria University, Edfina 22758, Egypt; asmaa.khafaga@alexu.edu.eg; 5Animal and Poultry Production Department, Faculty of Agriculture, Minia University, El-Minya 61519, Egypt; hamadaelwan83@mu.edu.eg; 6Department of Aquaculture and Biotechnology, Faculty of Aquaculture and Marine Fisheries, Arish University, Arish 45511, Egypt; m_fathy8789@yahoo.com; 7Arid Lands Cultivation Research Institute, City of Scientific Research and Technological Applications, New Borg El Arab, Alexandria 21934, Egypt; elsayed_hafez@yahoo.com

**Keywords:** pathogen, immune, biochemical, gene expression, aquaculture, tilapia, seabass, vibrio, *Streptococcus*

## Abstract

Nile tilapia (*Oreochromis niloticus*) and European sea bass (*Dicentrarchus labrax*) are economically significant species in Mediterranean countries, serving essential roles in the aquaculture industry due to high market demand and nutritional value. They experience substantial losses from bacterial pathogens *Vibrio anguillarum* and *Streptococcus iniae*, particularly at the onset of the summer season. The immune mechanisms involved in fish infections by *V. anguillarum* and *S. iniae* remain poorly understood. This study investigated their impact through experiments with control and *V. anguillarum*- and *S. iniae*-infected groups for each species. Blood samples were collected at 1, 3, and 7 days post bacterial injection to assess biochemical and immunological parameters, including enzyme activities (AST and ALT), oxidative markers (SOD, GPX, CAT, and MDA), and leukocyte counts. Further analyses included phagocyte activity, lysozyme activity, IgM levels, and complement C3 and C4 levels. Muscle tissues were sampled at 1, 3, and 7 days post injection to assess mRNA expression levels of 18 immune-relevant genes. The focus was on cytokines and immune-related genes, including pro-inflammatory cytokines (*TNF-α*, *TNF-β*, *IL-2*, *IL-6*, *IL-8*, *IL-12*, and *IFN-γ*), major histocompatibility complex components (*MHC-IIα* and *MHC-IIβ*), cytokine receptors (*CXCL-10* and *CD4-L2*), antimicrobial peptides (*Pleurocidin* and *β-defensin*), immune regulatory peptides (*Thymosin β12*, *Leap 2*, and *Lysozyme g*), and Galectins (*Galectin-8* and *Galectin-9*). *β-actin* was used as the housekeeping gene for normalization. Significant species-specific responses were observed in N. Tilapia and E. Sea Bass when infected with *V. anguillarum* and *S. iniae*, highlighting differences in biochemical, immune, and gene expression profiles. Notably, in N. Tilapia, AST levels significantly increased by day 7 during *S. iniae* infection, reaching 45.00 ± 3.00 (*p* < 0.05), indicating late-stage acute stress or tissue damage. Conversely, E. Sea Bass exhibited a significant rise in ALT levels by day 7 in the *S. iniae* group, peaking at 33.5 ± 3.20 (*p* < 0.05), suggesting liver distress or a systemic inflammatory response. On the immunological front, N. Tilapia showed significant increases in respiratory burst activity on day 1 for both pathogens, with values of 0.28 ± 0.03 for *V. anguillarum* and 0.25 ± 0.02 for *S. iniae* (*p* < 0.05), indicating robust initial immune activation. Finally, the gene expression analysis revealed a pronounced peak of *TNF-α* in E. Sea Bass by day 7 post *V. anguillarum* infection with a fold change of 6.120, suggesting a strong species-specific pro-inflammatory response strategy. Understanding these responses provides critical insights for enhancing disease management and productivity in aquaculture operations.

## 1. Introduction

Nile Tilapia (*Oreochromis niloticus*) is a pivotal freshwater species in Mediterranean countries’ aquaculture industry, especially Egypt, and is renowned for its adaptability and rapid growth rates. However, this species is increasingly vulnerable to bacterial infections, notably by *Vibrio anguillarum* and *Streptococcus iniae*, particularly during the onset of the summer months when water temperatures rise, creating optimal conditions for these pathogens to thrive. These bacterial pathogens lead to severe outbreaks, resulting in substantial economic losses due to increased mortality rates [1,2,3,4,5]. Globally, N. Tilapia ranks second only to carp in terms of aquaculture production, underscoring its critical role in food security and economic stability in aquaculture-dependent regions. The species is integral not only for local consumption but also as a significant export commodity.

Similarly, the European sea bass (*Dicentrarchus labrax*) holds substantial commercial significance, particularly in the Mediterranean basin [5,6]. It is a species highly prized for its market value and is central to regional aquaculture endeavors [7,8]. In 2022, Egypt’s aquaculture production reached 1.1 million metric tons, marking a slight decline from 1.6 million metric tons in 2020, with N. Tilapia contributing 61% of the national fish supply and 81% of African Tilapia production. Meanwhile, Egypt produced approximately 30,313 metric tons of E. Sea Bass in 2021, accounting for 11.52% of global production, making it a leading producer alongside Turkey and Greece (GAIN 2022, African Union 2023, FAO 2024). This prominence highlights the country’s strategic importance in the global aquaculture sector [9].

Both *O. niloticus* and *D. labrax* are essential to their respective aquaculture industries [5,10,11,12], yet their vulnerability to a spectrum of bacterial pathogens poses a persistent challenge. Addressing these health issues through improved management practices, targeted therapies, and enhanced biosecurity measures is crucial for sustaining their production and mitigating economic impacts [13,14]. This is especially pertinent as the demand for these species continues to grow amidst global efforts to meet rising demands for sustainable seafood [10],

Despite these advantages, the species are heavily impacted by bacterial disease outbreaks, which represent a significant barrier to the economic advancement of aquaculture in the region. These outbreaks can lead to alarming mortality rates [15]. One of the primary culprits is *V. anguillarum*—a Gram-negative pathogen known for causing vibriosis, which is characterized by skin ulcers and hemorrhagic septicemia in infected fish. Similarly, *S. iniae* is a Gram-positive bacterium that causes streptococcosis, leading to considerable economic crises in global aquaculture, notably affecting freshwater and marine species. This pathogen is recognized for inducing high mortality rates and manifests through symptoms such as meningoencephalitis and skin lesions, particularly under stress conditions [1,16,17,18].

The pathogenic mechanisms of *S. iniae* include virulence factors that enable efficient colonization and invasion of host tissues, contributing to its potent impact on aquaculture species [19,20]. Effective responses to *V. anguillarum* hinge on understanding its pathogenicity, which involves iron acquisition systems and toxin production that are key to its survival and pathogenicity in aquatic environments [1,2,3,4,5]. In fish, immune responses to bacterial invasions heavily involve organs such as the tissues muscles, kidney, and spleen, which are essential for both innate and adaptive immunity [21,22,23]. White blood cells play a crucial role, expressing immune-related genes crucial for both immediate and long-term immune responses [24].

Leukocytes, including neutrophils, monocytes/macrophages, and B lymphocytes, engage in robust intracellular and extracellular antimicrobial defenses [24], largely mediated by respiratory burst activities that produce reactive oxygen species (REOS) like superoxide anions, hydrogen peroxide, and hydroxyl radicals to attack pathogens. These responses are linked with the release of inflammatory cytokines that facilitate chemotaxis and mobilization of phagocytes, which are key to sustaining the fish’s immune defense [25,26].

This complex interplay between effector cells and pathogen-driven immune modulation underscores the need for targeted interventions in aquaculture management to mitigate the impacts of these bacterial pathogens, ensuring the sustainability and economic viability of fish farming in Egypt and beyond [7,27]. In this aspect, the discovery and understanding of defense genes that confer resistance against bacterial pathogens like *V. anguillarum* [1,2,3,4,5] and *S. iniae* [19,20] are crucial for developing strategies to safeguard fish such as *O. niloticus* and *D. labrax*. Key immune response genes, including cytokines and signaling molecules, play vital roles in orchestrating effective defense mechanisms [7,27].

In this regard, tumor necrosis factors (TNFs) are versatile cytokines that play crucial roles in maintaining health and contributing to disease development. Recent research highlights their involvement in modulating adaptive immune responses. However, the understanding of TNFs’ role in regulating adaptive immunity in early vertebrates remains limited. Recently, two TNF isoforms, *TNF-α* and *TNF-β*, were discovered in *O. niloticus*. By examining their sequence characteristics, researchers explored their regulatory impact on fish species’ adaptive immune response. Also, *TNF-α* and *TNF-β* are vital for inflammation and apoptosis, contributing to clearing infections [28].

Also, interleukins like *IL-2*, *IL-6*, *IL-8*, and *IL-12* are significant modulators of the immune system: *IL-2* promotes T-cell proliferation, *IL-6* acts as a pro-inflammatory cytokine, *IL-8* recruits neutrophils to infection sites, and *IL-12* enhances the differentiation of T-helper cells [29,30,31,32].

Additionally, antimicrobial peptides (AMPs) serve as a primary defense mechanism against various pathogens in fish and are critical components of the innate immune system in teleost fish [33]. In recent studies, fish have been identified as a valuable source of AMPs with significant potential as antibiotic alternatives. Bahabadi et al. [34] highlighted these peptides’ potential antibiotic properties across various fish species. Bhat et al. [35] demonstrated that the novel peptide *KK12YW* exhibits strong antibacterial activity against several fish bacterial pathogens. Masso-Silva and Diamond [36] reviewed the broad-spectrum antimicrobial and immunomodulatory effects of AMPs from fish species including winter flounder and Moses sole. Fekih-Zaghbib et al. [37] identified *Chrysophsin-1* and *Oncorhyncin-I* as potent antimicrobial biomarkers in the mucus of *Sparus aurata*, *D. labrax*, and *Pagrus pagrus*. Additionally, Ferez-Puche et al. [38] discovered a novel *β-defensin* gene in *Sparus aurata*, underscoring its importance in immune response regulation. Also, zebrafish (*Danio rerio*) exhibit an upregulation of the *nk-lysin* gene when challenged with the spring viraemia of the carp virus (SVCV) [39]. In certain fish species, such as the mandarin fish, some AMPs have been identified, including *piscidin* [40], *lysozyme g* [41], *β-defensin* [42], *hepcidin* [43], and Galectin genes [44]. Piscidin is predominantly expressed in the spleen and kidney of mandarin fish. While lysozyme appears to play a role in defense against *A. hydrophila* [45], β-defensin presents inhibitory effects against bacterial strains like *A. hydrophila* and *Staphylococcus aureus* [42]. Hepcidin contributes to the inflammatory response [43], and both *Galectin-8* and *Galectin-9* are effective against various bacterial strains. The roles of these genes in fighting bacterial infections can vary even within different species [44].

Understanding and leveraging these genes can potentially lead to the enhancement of disease resistance in aquaculture species, providing pathways to genetically resilient lines and innovative treatments, ultimately securing the sustainability and productivity of the aquaculture sector amidst bacterial challenges.

This study aims to investigate the biochemical and immune response profiles in N. Tilapia (*O. niloticus*) and E. Sea Bass (*D. labrax*) during bacterial pathogen challenges, with a particular focus on the gene expression associated with immune responses. Previous studies, such as those by Elbahnaswy et al. [46], Tang et al. [29], Hal et al. [30], and Li et al. [32] have explored close aspects in several fish species with different bacteria species, finding that specific genes related to cytokines, chemokines, and antimicrobial peptides are differentially expressed in response to pathogen exposure. These findings suggest that understanding gene expression patterns is critical for comprehending the immune response dynamics in aquaculture species.

Building on this existing knowledge, our research posits the following hypotheses: (1) exposure to bacterial pathogens will result in distinct, species-specific changes in the expression of immune-related genes in *O. niloticus* and *D. labrax*; (2) these changes will correlate with measurable differences in biochemical markers of stress and immune function. By comprehensively analyzing both species under controlled pathogen challenges, this study seeks to uncover significant species-specific responses that could inform new management strategies and therapeutic interventions. These insights are vital for enhancing disease resistance in aquaculture, thereby ensuring economic viability and sustaining seafood supply amid rising global demands.

## 2. Results

### 2.1. Identification and PCR Confirmation

Based on the successful amplification of three target genes, we were able to confidently confirm the identification of the bacterial strains. For *V. anguillarum*, we successfully amplified a 439-base pair (bp) segment of the *empA* gene (Figure 1 and Appendix A), following the previously established protocols. This result indicates an accurate detection using the method described by Xiao et al. [47].

Additionally, our PCR-based assay for *S. iniae* demonstrated reliable amplification of two specific DNA segments: a 300 bp fragment from the *Sin*/ 16S rRNA gene (Figure 2 and Appendix A) and an 870 bp fragment from the *lctO* gene (Figure 3 and Appendix A). These results align with the methodologies of Zlotkin et al. [48] and Mata et al. [49] confirming the presence of the species through conserved genetic regions.

### 2.2. Biochemical Parameters and Oxidation Assays

#### 2.2.1. Biochemical Parameters

The biochemical indices reveal significant alterations between N. Tilapia and E. Sea Bass when infected by *V. anguillarum* and *S. iniae*, with key differences observed across the varying days of infection. In N. Tilapia, there was a marked reduction in ALT and AST levels on day 1 and day 3 in the infected groups compared to the control, indicating a diminished immune response post infection. By day 7, however, a notable increase in AST was observed particularly with *S. iniae* infection (45.00 ± 3.00), signifying a potential late-stage acute stress or widespread tissue damage. Contrastingly, urea levels significantly rose by day 3 in infected groups, notably with *V. anguillarum*, suggesting altered protein metabolism possibly due to stress-induced catabolic processes. Creatinine levels consistently showed minor fluctuations yet remained generally comparable aside from slight increases indicating mild renal implications due to infection stress (Table 1).

In E. Sea Bass, unlike N. Tilapia, the initial days post infection (day 1 and day 3) showed relatively stable biochemical indices with no significant differences in ALT and AST levels among the control and infected groups, reflecting possibly an early nonspecific immune response or high baseline resilience. Nonetheless, by day 7, there was a profound escalation in both ALT (up to 33.5 ± 3.20 in *S. iniae* group) and AST levels, indicative of liver distress or systemic inflammatory response to both bacterial strains. Urea and creatinine levels also rose significantly, denoting either an overload of nitrogenous waste products or dehydration impact associated with increased metabolic demand due to prolonged immune activity (Table 1).

These biochemical changes across different days of infection and between the two types of fish highlight varied immune responses. The early decrease and later surge in enzyme activities in N. Tilapia imply a potentially biphasic immune response, whereas the E. Sea Bass exhibited a delayed yet pronounced stress response by day 7. The results suggest that fish species differ in their physiological and biochemical coping strategies to bacterial infections, reflecting on their varied immune response capabilities. The escalations in both ALT and AST in latter stages align with inflammatory responses where tissue damage or metabolic distress necessitates heightened enzyme activity, indicating both innate and adaptive immunity engagement.

#### 2.2.2. Oxidation Assays

Concerning MDA, on day 1, MDA levels in N. Tilapia showed a slight decrease in both groups infected with *V. anguillarum* and *S. iniae* compared to the control group, with values of 8.11 ± 0.12 IU/I and 8.55 ± 0.20 IU/I, respectively. This decrease suggests an early oxidative stress response. In contrast, E. Sea Bass exhibited a significant reduction in MDA levels to 5.80 ± 0.5 IU/I for the *V. anguillarum* group and 6.00 ± 0.60 IU/I for the *S. iniae* group, indicating a more pronounced oxidative response compared to N. Tilapia (Table 2).

Regarding antioxidant enzymes (SOD, GPX, and CAT) responses, on day 3, the activities of SOD for N. Tilapia infected with *V. anguillarum* significantly increased to 1.91 ± 0.14 IU/I compared to the control, demonstrating an upregulated antioxidant defense. Similarly, E. Sea Bass also showed increased SOD levels, notably with *V. anguillarum* infection reaching 1.93 ± 0.26 IU/I. A comparable trend was observed in GPX and CAT activities, where N. Tilapia and E. Sea Bass exhibited elevations, especially in GPX levels, reflecting enhanced detoxification of hydrogen peroxide. GPX activity reached 5.02 ± 0.19 IU/I in N. Tilapia with *V. anguillarum* infection, suggesting heightened oxidative stress mitigation.

For immune impacts, by day 7, variations in enzyme activities and MDA levels illustrated the cumulative effect of prolonged infection. In both fish species, infected groups exposed to either bacterial pathogen showed intricate variations. Most notably, E. Sea Bass demonstrated a return towards baseline values, although with slight residual increases in antioxidant enzymes such as CAT, particularly after *V. anguillarum* infection, noted at 566 ± 9.23 IU/I. This pattern suggests a partial recovery of immune balance possibly due to adaptive immune responses. Serum MDA levels peaked in E. Sea Bass to 13.00 ± 0.40 IU/I following *S. iniae* infection, indicating sustained oxidative stress, while N. Tilapia maintained slightly varied responses. Such differences across species and infection types underscore the interplay between innate responses and pathogen specificity (Table 2).

Overall, the comparative analysis reveals a significant species-specific antioxidant response to bacterial infections. E. Sea Bass showed more dynamic changes with both pathogens, possibly reflecting a diverse immune capability or varying stress tolerance. These findings highlight the critical nature of timely antioxidant enzyme responses in mitigating bacterial-induced oxidative stress, providing insights necessary for enhancing aquaculture disease management strategies. Among the results, the notable MDA level reduction and subsequent enzyme activity increases in the sea bass are significant, illustrating key points of defense activation and potential resilience to infections.

### 2.3. Total and Differential Leukocyte Counts

On day 1, N. Tilapia exposed to *V. anguillarum* and *S. iniae* displayed a significant increase in leukocytic counts compared to the control group, with total counts of 35.45 ± 2.78 and 31.40 ± 3.17, respectively (*p* < 0.05). This increase was accompanied by elevated lymphocytes, neutrophils, and monocytes, indicating a robust initial immune response to bacterial stress (Table 3). By day 3, there was a notable decrease in total leukocytic count, particularly in the *V. anguillarum* group, which fell to 21.70 ± 2.35, possibly suggesting an adaptive immune regulation or leukocyte redistribution (*p* < 0.05). By day 7, leukocytic counts surged again, reflecting an active immune response, with counts for *V. anguillarum* and *S. iniae* reaching 35.92 ± 3.35 and 35.20 ± 2.22, respectively, highlighting an adaptive immune phase (*p* < 0.05).

For European sea bass, a comparable pattern was observed. Initial infection with *V. anguillarum* and *S. iniae* on day 1 resulted in significant increases in leukocytic counts (35.20 ± 3.00 and 33.00 ± 3.20, respectively), demonstrating a heightened immune response (*p* < 0.05). Similar to N. Tilapia, a decrease was observed by day 3, particularly for *V. anguillarum*, where counts decreased to 21.00 ± 2.40 (*p* < 0.05). By day 7, however, there was a renewed increase in leukocyte activity, with *V. anguillarum* reaching a heightened count of 38.00 ± 3.60, emphasizing the protracted immune engagement in response to an ongoing pathogenic challenge (*p* < 0.05). This suggests the immune system’s dynamic capability to adjust its response over time, managing infection while maintaining homeostasis (Table 3).

### 2.4. Immunological Parameters

In N. Tilapia, the immune response parameters following infection with *V. anguillarum* and *S. iniae* demonstrate notable changes compared to the control group. On day 1, respiratory burst activity was significantly heightened in infected groups, measuring 0.28 ± 0.03 and 0.25 ± 0.02 O.D. at 630 nm for *V. anguillarum* and *S. iniae*, respectively, compared to 0.20 ± 0.02 in the control (*p* < 0.05). Serum lysozyme activity and immunoglobulin M levels also increased significantly in response to infection, indicating an active immune defense (*p* < 0.05). By day 3, these parameters remained elevated with a notable peak in respiratory burst activity for *V. anguillarum* at 0.33 ± 0.04, suggesting a sustained defensive state. On day 7, a reduction in some parameters like complement C3 and C4 was observed, with drops to 3.00 ± 0.50 and 1.80 ± 0.20 mg/dl, respectively, in *V. anguillarum* infected groups, indicating potential immune adaptation over time.

In E. Sea Bass, similar trends were observed post infection. On day 1, the respiratory burst activity increased to 0.30 ± 0.04 and 0.27 ± 0.03 for *V. anguillarum* and *S. iniae*, respectively, from a baseline of 0.22 ± 0.02 in controls (*p* < 0.05). Furthermore, serum lysozyme activity and immunoglobulin M levels exhibited significant enhancements, reflecting the activation of immune mechanisms (*p* < 0.05). By day 3, the highest respiratory burst activity was recorded in the *V. anguillarum* group at 0.35 ± 0.05, while sustained elevation in lysozyme activity was noted, indicating an ongoing immune response. By day 7, reductions in complement C3 and C4 levels were similar to those seen in N. Tilapia, with values dropping to 3.20 ± 0.70 and 1.90 ± 0.25 mg/dL in *V. anguillarum*-infected E. Sea Bass, suggesting modulation of the immune response as the fish manage chronic exposure to pathogens (Table 4).

### 2.5. Gene Expression Profiles in Response to Bacterial Infections in N. Tilapia and E. Sea Bass

#### 2.5.1. *TNF-α* and *TNF-β* Expression Dynamics and Response Comparisons in Analyses

*TNF-α* expression, which was notably temporal, and interspecies differences were observed. In N. Tilapia, *TNF-α* expression increased significantly from day 1 to day 7, with *V. anguillarum* infection yielding a fold change of 5.601 compared to *S. iniae*’s slightly higher induction of 5.956. These findings underscore the sustained pro-inflammatory response elicited by both bacterial pathogens. In contrast, E. Sea Bass exhibited a pronounced differential response with *V. anguillarum* infection showing a peak fold change of 6.120 by day 7, indicative of the species-specific innate immune strategies employed. Such data suggest the potential for differential regulation of inflammatory pathways between the two species (Figure 4A).

The expression of *TNF-β* revealed contrasting dynamics, particularly noticeable in the temporal responses to the infecting bacteria. While N. Tilapia showed a substantial increase by day 7 following *V. anguillarum* infection (2.703), *S. iniae* exposure led to a reduction to 0.656, highlighting pathogen-specific alterations in immune modulation. European sea bass, however, maintained elevated *TNF-β* levels under *V. anguillarum* influence, indicating a more consistent pro-inflammatory response across infection stages, thereby suggesting differences in cytokine regulatory mechanisms when compared to N. Tilapia (Figure 4B).

#### 2.5.2. *IL-2*, *IL-6*, *IL-8*, and *IL-12* Expression Patterns and Chemokine and Cytokine Perspectives

*IL-2* showed modest expression enhancements, with N. Tilapia responding more prominently by day 3 to *V. anguillarum* with a fold change of 1.422, implying a role in adaptive immune activation (Figure 4C). Meanwhile, *IL-6* responses were more prominent in both species, with N. Tilapia reaching a substantial fold change of 2.700 by day 7 against *S. iniae*, suggesting *IL-6*’s pivotal role in orchestrating prolonged inflammatory responses. This pattern emphasizes the differential latency and intensity of cytokine responses inherent to both host species and pathogen types (Figure 4D).

*IL-8* expression increased slightly in both fish species, particularly after *V. anguillarum* infection, reinforcing its role as a chemotactic agent in immune response initiation (Figure 4E). Conversely, *IL-12* expression exhibited a downward trend, which was more pronounced in N. Tilapia by day 7 (0.510 fold change post *V. anguillarum*), suggesting a downregulation mechanism or possible immune tolerance onset, thus differentially modulating the immune response over time (Figure 4F).

#### 2.5.3. *IFN-γ* and *MHC-II* Complex Interactions

*IFN-γ* displayed robust expression increases by day 7 across both bacterial challenges in both species, with E. Sea Bass shown a notable peak of 3.899 upon *V. anguillarum* infection, reflecting its integral role in activating cellular immune responses and enhancing macrophage activity (Figure 5A). Post-infection, *MHC-IIα* and *MHC-IIβ* genes experienced marked downregulation by day 7, particularly in N. Tilapia following *V. anguillarum* (*MHC-IIα* dropping to 0.310), indicating differential antigen processing route adaptations between the two species (Figure 5B,C).

#### 2.5.4. Immunologically Important Molecules

*CXCL-10* expression depicted an upward trajectory, especially post *S. iniae* infection, reaching 7.158 in N. Tilapia on day 7 and demonstrating its critical role in leukocyte migration and positioning (Figure 5D). Concerning *CD4-12*, on day 3, exposure to *S. iniae* resulted in a significant upregulation of gene expression in both groups, with expression levels reaching 2.712 ± 0.050 and 4.234 ± 0.060, respectively, highlighting the differential response to treatment conditions (Figure 5E). On the other side, *Pleurocidin* expression decreased over time, possibly pointing to its involvement in immediate innate responses rather than sustained immune activity (Figure 5F).

Antimicrobial peptides, including *Thymosin β12* (Figure 6A) and *lysozyme g* (Figure 6B), were upregulated, underscoring their roles in immediate pathogen neutralization. Regarding the *Leap 2* gene, exposure to *S. iniae* resulted the in significant upregulation of gene expression in both treatment groups. On day 1, the E. Sea Bass infected with *S. iniae* exhibited an expression level of 11.340 ± 0.220, while on day 7, the N. Tilapia infected with *S. iniae* reached 6.523 ± 0.100, indicating a strong and sustained immune response (Figure 6C). In the case of the *β-defensin* gene, exposure to *V. anguillarum* in N. Tilapia induced significant upregulation, particularly on day 3, with expression levels reaching 3.599 ± 0.070. Conversely, *S. iniae* exposure in N. Tilapia resulted in the highest expression on day 1, with levels of 2.403 ± 0.050 demonstrating varied temporal expression patterns in response to different pathogens (Figure 6D).

#### 2.5.5. Galectin-Mediated Immune Modulation

The upregulation of *Galectins 8* and *9*, notably in E. Sea Bass at various time points, particularly following *S. iniae* infection, emphasizes their involvement in modulating leukocyte activity and apoptosis, with fold changes peaking at 16.780 for *Galectin-9* on day 1. This suggests a significant role in cell signaling during immune defense, highlighting pathogen- and host-specific regulatory pathways leveraged during bacterial invasions (Figure 6E,F).

These findings delineate the immunological responses of N. Tilapia and E. Sea Bass to infections by *V. anguillarum* and *S. iniae*, highlighting distinct species-specific defense mechanisms and temporal patterns in gene expression. Notably, N. Tilapia exhibited strong pro-inflammatory responses, especially through *TNF-α* and *IFN-γ*, suggesting a robust innate response mechanism, whereas E. Sea Bass displayed heightened regulation and modulation through cytokine activities, indicating a more balanced immune strategy. The variability in their immune responses underscores the necessity of tailored approaches in managing fish health within aquaculture systems.

## 3. Discussion

### 3.1. Bacterial Infection Challenges

In recent years, advancing our understanding of the immune responses in different fish species during bacterial challenges has become imperative, especially given the economic impact of aquaculture. The studies outlined herein provide a comprehensive view of the intricate interplay between fish immune systems and pathogenic bacteria, offering insights that are crucial for developing targeted vaccination strategies [1,17,50,51].

The comprehensive evaluation of biochemical parameters, oxidation assays, and differential leukocyte counts, alongside immunological parameters and gene expression profiles, provides a multifaceted understanding of the immune response in N. Tilapia (*O. niloticus*) and E. Sea Bass (*D. labrax*) during pathogen challenges and infections. Oxidation assays are crucial for assessing the oxidative stress levels in these fish species, highlighting how their physiological systems combat oxidative damage caused by pathogens [52,53,54]. Meanwhile, total and differential leukocyte counts offer insights into the adaptability and mobilization of immune cells in response to infections, which is pivotal for understanding the overall immune readiness and resilience of the organisms [21,55,56]. Immunological parameters, such as the levels of cytokines and other immune mediators, further inform about the signaling pathways activated during infections, revealing key components of the host defense strategy [21,57]. Additionally, analyzing gene expression profiles offers a molecular perspective on how specific genes are upregulated or downregulated during bacterial infections, enabling researchers to pinpoint critical genes involved in immune response pathways [58,59,60]. Together, these measurements not only elucidate the complex interplay of immune components in these fish species during pathogen exposure but also facilitate the identification of potential biomarkers and targets for enhancing disease resistance through selective breeding or therapeutic interventions [7,27]. Fortunately, in the present study, we provided a comprehensive investigation on two important fish species, *O. niloticus* and *D. labrax*, which are among the most significant fish in Mediterranean countries.

The current study underscores the critical need for advancing our understanding of the immune response in fish, particularly N. Tilapia and E. Sea Bass, as they suffer significant losses due to infections by poorly understood bacterial pathogens like *V. anguillarum* and *S. iniae* during the summer season. The present findings revealed that N. Tilapia experienced early immune suppression and later tissue damage with bacterial infections, whereas E. Sea Bass show initial resilience followed by a strong inflammatory response.

In this regard, Manchanayake et al. [50] reported that *V. anguillarum*-related diseases are of global importance, affecting not only marine aquaculture systems but also wild fish populations. Vibriosis results in considerable economic losses for fish farming operations. Meanwhile, Xiong et al. [20] confirmed that *S. iniae* is a significant fish pathogen impacting the aquaculture industry. However, no comparative analyses of aquaculture infections have been conducted to date.

Integrating the findings of current immune response studies on N. Tilapia and E. Sea Bass with comprehensive reviews by Frans et al. [61] and Ina-Salwany et al. [62] highlights the pressing need for effective disease management and innovative vaccination strategies to mitigate the impacts of pathogens like *V. anguillarum* and *S. iniae* in aquaculture, underscoring a collaborative approach to enhance resilience against these bacterial threats.

Furthering our understanding of bacterial diseases in aquaculture, Frans et al. [61] and Ina-Salwany et al. [62] reviewed the impacts of vibriosis, caused by Vibrio pathogens, detailing the virulence factors and current prevention strategies. These reviews side by side with our findings emphasize the importance of comprehensive disease management practices in mitigating the widespread economic losses caused by vibriosis in marine and freshwater aquaculture.

In this aspect, diverse pathogen challenges in different fish species, such as the Japanese flounder in the study by Sun et al. [63], demonstrate how DNA vaccines against S. iniae and *V. anguillarum* can lead to robust bivalent immunity. This highlights the potential of such vaccines in enhancing cross-protection capabilities. Additionally, Ahangarzadeh et al. [15] provided promising results using a killed polyvalent vaccine in Asian Sea Bass, achieving significant improvements in survival rates, which could inform future vaccine developments for other fish species including N. Tilapia and E. Sea Bass.

In the context of N. Tilapia, studies such as those conducted by Shoemaker et al. [51] and Wang et al. [64] strengthen the case for bivalent and ghost vaccines, respectively. These demonstrate improved immune responses and survival following pathogen challenges, offering a feasible approach to combat concurrent infections by *S. iniae* and various Vibrio species. The potential for genetic improvements, as discussed by Vela-Avitúa et al. [65], shows that marker-assisted selection can be a tangible method to breed fish with enhanced disease resistance, opening new avenues for sustainable aquaculture. On the other side, a study by Hal et al. [30] revealed that infections by *Aeromonas hydrophila* and Pseudomonas fluorescens lead to pronounced clinical symptoms and high mortality rates in N. Tilapia, with signs such as skin hemorrhages, swollen abdomens, and eye cloudiness. The infections also triggered severe damage to vital organs like the liver and kidneys, reflecting the struggle of the fish to combat the bacterial onslaught, which underscores the vulnerability of N. Tilapia to these pathogens [66].

Our study aimed to further delineate these responses by examining the expression and immune profiles of these species under controlled pathogen infections. The focus was not only on established pathogens but also on exploring biochemical parameters, oxidation assays, differential leukocyte counts, immunological parameters, and gene expression. This integration of immunological, genetic, and pathological insights is crucial for developing holistic strategies to manage fish health in aquaculture.

### 3.2. Biochemical Parameters

Biochemical indices serve as important diagnostic tools to assess the health status of fish in aquaculture systems [55]. The liver, a critical organ for detoxification and metabolism, is evaluated using plasma transaminase enzymes, ALT and AST, which are key in cellular nitrogen metabolism. Elevated levels of these enzymes may suggest liver cell damage due to toxic exposure [67].

In this study, a significant reduction in ALT and AST activities was observed in infected N. Tilapia on days 1 and 3 when compared to the control group, suggesting an initial suppression of the immune response following infection. By day 7, AST levels increased notably in groups infected with *S. iniae*, possibly indicating acute stress or extensive tissue damage at this later stage. This pattern echoes the research by Ali et al. [68] where boric acid effectively reduced systemic tissue damage in N. Tilapia infected with Saprolegnia parasitica, as indicated by decreased ALT and AST levels. Similarly, Rastiannasab et al. [69] found increased AST and ALT levels in common carp (*Cyprinus carpio*) infected with parasites (*Dactylogyrus* spp. and *Gyrodactylus* spp.), highlighting the impact of infections on liver and kidney function and underlining the importance of biochemical markers in assessing infection severity.

In European sea bass, ALT and AST levels remained stable on days 1 and 3 post infection across both the control and infected groups, possibly indicating a strong early immune response or innate resilience to the bacterial challenge. By day 7, significant increases in these enzyme levels suggested liver stress or systemic inflammation due to persistent infection. These findings align with those of the study by Rahimikia et al. [70] where exposure to nickel in goldfish (*Carassius auratus*) led to increased activities of antioxidant enzymes such as SOD and GPx as well as elevated hepatic enzymes like AST and ALT, indicating a physiological response to chemical stress rather than infection.

Nitrogen metabolism products, such as BUN and creatinine, are typically evaluated in the blood of various organisms, including fish. The BUN levels tend to be consistent across different species of freshwater fish [71]. Creatinine, which is produced from the metabolism of creatine in the muscles, is eliminated via the kidneys [72]. Therefore, elevated creatinine levels in the blood of fish could indicate muscle damage or renal issues that hinder its removal. In N. Tilapia, urea levels rose significantly by day 3, especially in *V. anguillarum*-infected groups, possibly due to protein metabolism shifts under stress. Creatinine levels showed minor fluctuations but generally remained comparable, indicating mild renal implications due to infection-related stress. These observations are consistent with those from the work of Clark et al. [73] who investigated similar metabolic disturbances in rainbow trout during bacterial infections. They highlighted that such infections led to alterations in the urea cycle, which could contribute to the observed increases in urea levels. Like our results, Clark et al. findings suggest that bacterial infections can impact nitrogen metabolism, manifesting as changes in key metabolic products such as urea and creatinine. These consistent trends underline the importance of monitoring these nitrogen metabolites to understand the physiological impacts of infections on fish.

E. Sea Bass, however, exhibited significant rises in both urea and creatinine by day 7, likely reflecting increased metabolic demands and nitrogenous waste. This scenario corresponds with those in studies by Senthamarai et al. [74], which reported significant urea and creatinine elevations in fish under metabolic stress. Infectious diseases raise metabolic rates and nitrogenous waste in aquaculture, as seen in our E. Sea Bass results. Elevated creatinine suggests renal or muscle issues, highlighting the need to monitor these changes for better infection management.

These findings underscore the varied immune strategies and biochemical responses of N. Tilapia and E. Sea Bass to bacterial infections, highlighting species-specific physiological adaptations. The observed changes in enzyme activities suggest both innate and adaptive immune involvement, with different responses at varying stages of infection [22,75,76,77]. This enhancement could be attributed to its function in disrupting the chain of free radical reactions, allowing free radicals to extract hydrogen atoms from antioxidant molecules rather than from polyunsaturated fatty acids [78,79]. As a result, less reactive radical species are formed, which help safeguard kidney tissue against peroxidative damage [7,53,80,81,82].

### 3.3. Oxidation Assays

The generation of ROS and the release of free radicals lead to oxidative stress. Fish possess antioxidant enzymes like GPX, SOD, and CAT, which protect against ROS by reducing their production and safeguarding cellular structures and metabolic functions from harm [83]. SOD and CAT play a crucial role in breaking down ROS and minimizing lipid peroxidation, which are assessed by measuring MDA levels [84,85,86]. Failure to neutralize free radicals can result in DNA damage and cell death [86]. In the current study, E. Sea Bass demonstrated significantly higher SOD, GPX, and CAT activities compared to N. Tilapia, with a notable reduction in MDA levels. By day 3, both species showed increased antioxidant enzyme activities, but E. Sea Bass exhibited a stronger oxidative response and signs of immunological recovery by day 7, even under the stress of *S. iniae* infections. This pronounced enzyme activity in E. Sea Bass highlights key defensive adaptations to bacterial challenges and underscores the importance of enhancing antioxidant defenses in aquaculture to better manage infections.

### 3.4. Total and Differential Leukocyte Counts

The study of leukocyte counts reveals key immune responses in N. Tilapia and E. Sea Bass when infected with *V. anguillarum* and *S. iniae*. Both species showed a strong immune reaction on day 1, with elevated leukocyte counts. By day 3, counts decreased, particularly with *V. anguillarum*, indicating regulatory adaptation. By day 7, counts increased again, suggesting an adaptive immune phase. These patterns underscore the dynamic nature of the fish immune system, balancing infection control with homeostasis. The findings emphasize the importance of adaptive immunity in aquaculture management.

The studies on leukocyte responses in fish provide valuable insights into the immune dynamics of aquatic species facing bacterial infections. Research on N. Tilapia infected with bacteria like *Enterococcus* sp. and *Flavobacterium columnare* showed considerable shifts in leukocyte profiles, including increased lymphocytes and neutrophils, indicating robust initial immune reactions [87,88]. Similarly, studies on carp (*Cyprinus carpio*) infected with *Aeromonas salmonicida* demonstrated significant increases in total leukocyte counts and differential counts, pointing to dynamic immune responses [89]. These observations are consistent with findings in African catfish (*Clarias gariepinus*) challenged with *Escherichia coli* and *Vibrio fischeri*, where hematological assessments revealed changes in leukocyte counts and other blood parameters, suggesting adaptive immune modulation [90]. Non-lethal methodologies developed for studying leukocyte subpopulations, as seen in barramundi (*Lates calcarifer*), further highlight the importance of precise hematological assessments for understanding and managing fish health in aquaculture [91]. The present study on total and differential leukocyte counts highlights notable immune responses in both N. Tilapia and E. Sea Bass when infected with *V. anguillarum* and *S. iniae*.

### 3.5. Immunological Parameters

This study highlights the significant immune activation in N. Tilapia and E. Sea Bass following infection with *V. anguillarum* and *S. iniae*. In N. Tilapia, respiratory burst activity increased notably on day 1, showing higher optical density values compared to controls, which indicates a strong initial immune response. This elevated activity continued, peaking by day 3 for *V. anguillarum*. Additionally, significant increases in serum lysozyme activity and IgM were observed, reinforcing the robust immune defense. These findings demonstrate the fish’s capacity to mount a dynamic immune response to infections, providing useful insights for enhancing aquaculture health management. A similar immune response was observed in *Solea senegalensis* infected with Photobacterium damselae subsp (Phdp) [92,93] and in blunt snout bream (*Megalobrama amblycephala*) on days 1, 3, 5, 14, and 21 following an *A. hydrophila* challenge [94]. Additionally, the expression of the lysozyme gene was notably increased in the liver of *S. senegalensis* infected with Phdp [95]. The increased serum lysozyme levels in infected fish serve as a natural defense mechanism, highlighting its role as a crucial enzyme in the innate immune system. Lysozymes possess lytic and opsonic activities, which are pivotal in activating the complement system and enhancing phagocytosis against Gram-negative bacteria [96].

IgM plays a crucial role in the immune responses of fish, both innate and adaptive, when facing bacterial infections [97]. It facilitates the activation of the complement system, which is essential for lysing and opsonizing pathogens [98]. An increase in IgM levels was observed in N. Tilapia infected with *A. hydrophila* [99], and similarly, elevated antibody levels were found in the serum of *Solea senegalensis* infected with Phdp [95]. Our findings demonstrate a significant increase in IgM levels in both N. Tilapia and E. Sea Bass following infections with *V. anguillarum* and *S. iniae*. On the first day, N. Tilapia infected with *V. anguillarum* showed noticeably higher IgM levels compared to the control group. Similarly, Tilapia infected with S. iniae displayed a substantial rise in IgM. By the third day, the *V. anguillarum*-infected group maintained these elevated levels, indicating a persistent immune response. E. Sea Bass exhibited a consistent pattern, with IgM levels significantly rising on the first day for both *V. anguillarum* and *S. iniae*. These increased IgM levels on the initial days suggest a robust immunological adaptation, likely enhancing the fish’s ability to effectively address bacterial challenges. This elevated serum IgM results from exposure to bacterial infections, which triggers a cascade of immunological reactions, leading to lymphocyte activation and subsequent IgM synthesis and secretion [100]. This increase may also relate to IgM’s role in mediating agglutination to promote phagocytosis and pathogen clearance [101].

The complement system serves as a crucial bridge between innate and adaptive immunity in vertebrates and invertebrates, providing defense against pathogen invasion [98,102]. Key components of this system, C3 and C4, are associated with the alpha-2 macroglobulin superfamily of proteins that contain thioesters [103]. C3 undergoes activation through three principal pathways—classical, alternative, and lectin—resulting in its division into the fragments C3a and C3b [104]. In contrast, C4 plays a pivotal role in both the classical and lectin pathways by forming C3 and C5 convertases [98,105]. Through complement-mediated opsonization, C3 and/or C4 become covalently bound to microbial surfaces, facilitating their recognition and subsequent phagocytosis by phagocytes equipped with complement receptors [106]. In the current study, by day 7, notable reductions in complement components C3 and C4 were observed, indicating an adaptive immune response developing over time. This trend aligns with observations in N. Tilapia and reflects immune modulation following chronic pathogen exposure. Additionally, E. Sea Bass exhibited an increase in respiratory burst activity on day 1, reaching levels of 0.30 ± 0.04 and 0.27 ± 0.03 for *V. anguillarum* and *S. iniae*, respectively (*p* < 0.05), along with heightened lysozyme activity and IgM levels. The peak respiratory burst was recorded at 0.35 ± 0.05 by day 3 in the *V. anguillarum* group. Several studies have demonstrated the activation of complement pathways in response to specific infections in fish species, including *M. amblycephala* challenged by *A. hydrophila*, mandarin fish (*Siniperca chuatsi*) infected with Flavobacterium columnare, and soiny mullet (*Liza haematocheila*) in response to *Streptococcus dysgalactiae* [107,108,109]. This pronounced activation underscores the essential role of the complement system in fish immune responses, encompassing functions such as bacterial lysis via membrane attack complexes, opsonization enhancing phagocytosis, inflammatory responses, and the release of anaphylatoxins C3a and C5a, which recruit granulocytes to infection sites, thus bolstering host defenses [104,110].

Comparatively, the study by Elbahnaswy et al. [46] noted higher respiratory bursts in *A. hydrophila* throughout and in *P. damselae* on days 3 and 7. IgM levels increased in both groups on days 1 and 7. Complement C4 was elevated in *A. hydrophila* across the period but only on day 7 in *P. damselae*. These differences underscore distinct immune responses to each pathogen.

### 3.6. Gene Expression

In the current study, gene expression in muscle samples was examined to understand immune responses in fish. Regarding this aspect, the studies by Chapela et al. [111] and Valenzuela et al. [112] both highlight the vital role of examining muscle tissues in advancing the understanding of immune responses in fish. Chapela et al. [111] developed a multiplex quantitative PCR (qPCR) method for detecting bacterial infections—specifically Lactococcus garvieae, Yersinia ruckeri, and Flavobacterium psychrophilum in rainbow trout (Oncorhynchus mykiss). This method demonstrated 100% relative sensitivity and high specificity, proving to be both reliable and efficient with an accuracy range of 97.5% to 108.8%, which is beneficial for rapid diagnostics in aquaculture. On the other hand, Valenzuela et al. [112] focused on immune responses within the skeletal muscles of fine flounder (Paralichthys adspersus) challenged with Vibrio ordalii. Their findings showcased the skeletal muscle as a crucial immunological organ, indicated by the upregulation of immune-related genes and activation of key pathways like NFĸB and P38-MAPK/AP-1 along with the production of antimicrobial peptides such as hepcidin and Leap-2. Both studies resonate with our research, which also employs a qRT-PCR to analyze differential immune gene responses in muscle tissue, thus contributing to a broader comprehension of innate immune mechanisms in fish [111,112].

#### 3.6.1. *TNF-α* and *TNF-β* Expression Dynamics and Response Comparisons in Analyses

In the present study, the investigation into *TNF-α* and *TNF-β* expression dynamics across N. Tilapia and E. Sea Bass provides critical insights into interspecies variations in immune responses to bacterial infections. In N. Tilapia, *TNF-α* displays a gradual escalation from day 1 to day 7 for both *V. anguillarum* and *S. iniae*, suggesting a sustained pro-inflammatory reaction irrespective of the pathogen. This indicates tilapia’s robust inflammatory mechanisms targeting diverse pathogens. In contrast, E. Sea Bass exhibits a distinct immune strategy, with *V. anguillarum* infection provoking a peak *TNF-α* response by day 7, highlighting species-specific differences in cytokine-mediated inflammation. The *TNF-β* expression unveils further complexity; while tilapia experiences a strong response to *V. anguillarum* and a downturn with *S. iniae*, the steady *TNF-β* levels in E. Sea Bass suggest divergent cytokine regulatory pathways. This highlights how distinct evolutionary paths have influenced immune strategies in these species.

In this aspect, Li et al. [28] reported that TNFs are pivotal to maintaining health and managing disease, with emerging studies highlighting their significance in regulating adaptive immunity. Despite this recognition, their role in the immune responses of early vertebrates is not thoroughly understood. Their study identified two TNF isoforms, *TNF-α* and *TNF-β*, in N. Tilapia and investigated their function within the species’ immune system. Both *TNF-α* and *TNF-β* are evolutionarily conserved and exhibit high expression levels in the gills, suggesting a critical physiological role. In response to infection with *Streptococcus agalactiae*, these isoforms are upregulated in spleen lymphocytes, underscoring their importance in adaptive immune responses. Additionally, the isoforms can induce apoptosis in leukocytes, with *TNF-β* specifically activating *Caspase-8*, indicating distinct pathways in immune regulation.

#### 3.6.2. *IL-2*, *IL-6*, *IL-8*, and *IL-12* Expression Patterns and Chemokine and Cytokine Perspectives

This study explores interleukin dynamics in *N. Tilapia*: *IL-2* modestly enhances adaptive immunity, *IL-6* is pivotal against *S. iniae*-induced inflammation, and *IL-8* slightly rises post *V. anguillarum* infection, confirming its chemotactic role, while *IL-12* decreases, indicating immune tolerance mechanisms. The findings are consistent with Elbahnaswy et al. [46], who researched interleukin gene expression in *N. Tilapia* post *A. hydrophila* and *P. damselae* challenges. On day 1, *IL-6* expression remained stable in the spleen, and *IL-8* and *IL-1β* were downregulated only after *P. damselae*. In the head kidney, significant *IL-8* downregulation occurred after *P. damselae*, with little change post bacterial challenge. By day 3, *IL-8* and *IL-6* were still downregulated in the spleen, while by day 7, *IL-8* expression was stable. This study highlights dynamic interleukin gene responses to infections in *N. Tilapia*.

Tang et al. [29] identified *IL-2* in *Paralichthys olivaceus*, showing its role in immune response. *IL-2* is expressed in leukocytes, spleen, and hindgut, increasing post *Edwardsiella tarda* and viral infections (HIRRV). An *IL-2* expression vector improved immune gene expression and flounder survival against HIRRV, underlining *IL-2*’s potential in flounder aquaculture. Hal et al. [30] studied *A. hydrophila* and *P. fluorescens* impact on *N. Tilapia* immune response, emphasizing changes in immune genes like *IL-1β*, hepcidin, and *CYP1A*. Wei et al. [31] analyzed the *IL-6* gene in *N. Tilapia* post bacterial infection, noting elevated *IL-6* expression upon LPS and *Streptococcus agalactiae* exposure, which is crucial for antibody production. Li et al. [32] found that *IL-8* is highly expressed in *N. Tilapia* post *S. agalactiae* and *Aeromonas hydrophila* infections, enhancing immune functions and inflammation. Wang et al. [113] explored *IL-8* as an adjuvant in *S. iniae* vaccinations in catfish, enhancing immune responses but noting limited long-term effects. Xiao et al. [114] identified *IL-8* and *IL-10* in *Pelteobagrus fulvidraco*, revealing their roles in immune response and affected by *Clostridium butyricum* and *Aeromonas punctata*.

Matsumoto et al. [115] studied *IL-12* regulation in *Seriola dumerili* against intracellular infection, showing its dependence on transcription factors *IRF-1* and *AP-1*. Heeb et al. [116] examined *IL-4* and *IL-13* evolutions, working through the *IL-4* receptor and regulating type 2 immunity. Wang et al. [117] identified key chemokine genes critical during Singapore grouper iridovirus infection, enhancing understanding of fish immunity.

#### 3.6.3. *IFN-γ* and *MHC-II* Complex Interactions

The interactions between *IFN-γ* and *MHC-II* reveal a complex immune landscape. *E. Sea Bass* exhibits significant *IFN-γ* increases after *V. anguillarum* infection, highlighting its role in cellular immunity, while *N. Tilapia* displays *MHC-II* downregulation post *V. anguillarum*, suggesting strategic antigen processing modulation. These findings agree with those of Elbahnaswy et al. [46], who studied *MHC-IIα*, *TNF-α*, *TLR-7*, and *NF-κB* gene expression in *N. Tilapia*. On day 1, minimal changes were noted in the spleen, except for downregulation of *NF-κB* after *P. damselae*. Significant downregulation of *TLR-7* and *NF-κB* was observed in the head kidney post *P. damselae*, continuing through day 3, especially after exposure to *A. hydrophila*. By day 7, *NF-κB* showed a notable decrease in the spleen, with *TLR-7* downregulation persisting. These results indicate diverse gene expression responses to bacterial pathogens. Yu et al. [118] report that *IFN-γ* enhances macrophage immune function, increasing their ability to engulf and kill *Mycobacterium marinum* via heightened phagocytosis, apoptosis, and reactive molecule production like H_2_O_2_ and NO, along with upregulated *TLR2* and *Caspase 8*. *IFN-γ* also boosts antigen presentation by elevating *MHC* molecule expression, balancing the immune response through cytokine modulation, Reith et al. [119] found *MHC-II* essential for adaptive immunity, aiding CD4+ T cell activation. Arsenite impacts *MHC-II* expression via oxidative stress, which is reversible by HDAC1/2 inhibition. Dimethyl fumarate disrupts *IFN-γ* response by altering transcriptional control, providing insights into *MHC-II*-associated diseases.

#### 3.6.4. Immunologically Important Molecules

Chemokines are crucial for immune system function, directing the movement and activation of leukocytes [120,121,122]. This study highlights significant increases in *CXCL-10* in *N. Tilapia* post *S. iniae* infection, emphasizing leukocyte recruitment, while decreased *Pleurocidin* expression suggests a role in rapid initial responses. Upregulation of antimicrobial peptides like *Thymosin β12* and *lysozyme g* underscores their defensive capabilities. Schott et al. [122] found that susceptibility to *Chlamydia trachomatis* involves pathogen-evading immune detection tactics, specifically suppressing *CXCL10* through CPAF, linked to SNP rs2869462 and suppressing *RANTES* via different mechanisms. Cambier et al. [120] noted that *CXCL8* attracts neutrophils and interacts with *CXCR1*, *CXCR2*, *ACKR1*, and glycosaminoglycans, making it a potential therapeutic target. In contrast, *CXCL12* maintains leukocytes in the bone marrow, which is vital for embryogenesis, hematopoiesis, and angiogenesis through *CXCR4*, *ACKR1*, and *ACKR3*. Understanding these chemokines opens therapeutic possibilities.

For *Thymosin β12*, Zhang et al. [123] identified its crucial role in antibacterial immunity in *Urechis unicinctus*, noting high expression in the body wall and increased expression post LPS injection sequentially in various organs. *Thymosin β12* effectively inhibits bacterial growth, highlighting its role in innate immunity in marine invertebrates.

Regarding *lysozyme g*, Song et al. [124] emphasized its role in fish immunity amid rising disease from ecological degradation. Safarian et al. [125] explored the molecular traits and antibacterial efficiency of the g-type lysozyme in *Euryglossa orientalis*, noting evolutionary significance and strong activity against fish pathogens and underscoring their potential in fish health management.

#### 3.6.5. Galectin-Mediated Immune Modulation

Finally, the surge in Galectins, notably in E. Sea bass when challenged by *S. iniae*, indicates their integral role in modulating immune functions like leukocyte activity and apoptosis. *Galectin-9*’s impressive increase suggests sophisticated galectin-mediated pathways leveraged during bacterial confrontations.

It is worth mentioning that the potential of fish AMPs as alternative therapies against pathogens has sparked significant interest [126]. For example, 277 potential AMP sequences have been identified from the zebrafish transcriptome, and 11 specific AMPs have been characterized [127,128]. Similarly, analyses of the channel catfish genome have revealed 605 possible AMP sequences, with 11 being recognized [129]. In certain fish species, such as the mandarin fish, some AMPs have been identified, including Piscidin [40], lysozyme g [41], β-defensin [42], hepcidin [43], and galectin proteins [44]. *Piscidin* is predominantly expressed in the spleen and kidney of mandarin fish. While lysozyme appears to play a role in defense against *A. hydrophila* [45], *β-defensin* presents inhibitory effects against bacterial strains like *A. hydrophila* and Staphylococcus aureus [42]. Hepcidin contributes to the inflammatory response [43], and both *Galectin-8* and *Galectin-9* are effective against various bacterial strains [44].

Overall, this comparative analysis highlights interspecies differences and the complexity of immune responses tailored to bacterial challenges. Understanding these dynamics extends our understanding of fish immunology, offering insights for enhancing disease resistance in aquaculture systems.

Briefly, this study sheds light on the distinct biochemical and immunological responses of N. Tilapia and E. Sea Bass when challenged with *V. anguillarum* and *S. iniae*. By elucidating the specific immune profiles and gene expression changes in these fish species, the findings provide critical insights into the underlying mechanisms of pathogen resistance. However, the research also underscores the necessity for comprehensive aquaculture management strategies that address environmental and operational factors contributing to disease outbreaks. Through improved aquaculture practices focused on stress reduction, optimal water quality, and biosecurity, the industry can mitigate the risk of bacterial infections and enhance fish survival and overall productivity.

It is worth mentioning that future research should investigate how environmental conditions in aquaculture systems impact their efficacy. Temperature changes, for instance, can stress fish, weakening their immune systems and increasing susceptibility to infections. Additionally, intensive aquaculture practices may create environments conducive to pathogen proliferation, leading to more frequent disease outbreaks. Understanding these dynamics can guide future strategies that integrate both biotic and abiotic factors, optimizing disease resistance and overall fish health. By addressing these variables, researchers can work towards more sustainable and resilient aquaculture systems that effectively balance biological potential with challenges posed by environmental factors. Future research efforts, therefore, should focus on exploring these environmental interactions to pave the way for adaptive management strategies that sustain the health of economically significant fish species.

## 4. Materials and Methods

### 4.1. Bacterial Strains, Identification and Conformation

Strains of *V. anguillarum* and *S. iniae* were isolated from the internal organs of *N. Tilapia* and *E. Sea bass* that exhibited symptoms of acute hemorrhagic septicemia, following the methodology described by Austin et al. [130]. The fish samples showing signs of infection were collected during the summer season (September), when warmer temperatures contribute to increased susceptibility to infections. These samples were obtained from a commercial fish farm located in the Kafrelsheikh governorate, Egypt.

To proceed with the analysis, the bacterial isolates were grown on Tryptone-Soya-Agar (TSA) plates or brain and heart infusion-agar (BHIA) for a duration of 24 h at a temperature of 28 °C. Subsequently, DNA was extracted utilizing the Bacteria DNA Isolation Mini Kit (Park, Nanjing, Vazyme, China), adhering strictly to the instructions provided by the manufacturer. This extracted DNA served as the template for further testing. Both positive and negative controls were employed in the procedure to ensure accuracy and validity, with the negative control functioning without any template DNA.

The identification of these strains was confirmed through PCR amplification of target DNA sequences and 16s rRNA sequencing, demonstrating a 100% match to each respective species. For *V. anguillarum* detection, a 439-base pair (bp) segment of the empA gene was amplified following the protocols outlined by Xiao et al. [47]. Additionally, two distinct DNA primers targeting conserved regions of the 16S rRNA gene (300 bp, *Sin*) and the lactate oxidase gene (870 bp, *lctO*) were utilized for a PCR-based assay to identify *S. iniae*, as previously described by Zlotkin et al. [48] and Mata et al. [49], respectively. Detailed information on the primers is provided in Table 5.

Regarding the PCR conditions, a 25 μL reaction mixture was prepared, which included 12.5 μL of the master mix (Toshima-ku, Tokyo, TaKaRa, Japan), 1 μL of template DNA (30 ng), 1 μL of each primer (10 pmol/μL), and sterile water to reach a final volume of 25 μL. The PCR was performed using a thermal cycler (T100 Thermal Cycler from BIO RAD, Singapore) under the following conditions: initial denaturation at 95 °C for 2 min, followed by 35 cycles of 94 °C for 1 min, annealing at 55~57 °C for 1 min, and extension at 72 °C for 1 min with a final extension at 72 °C for 7 min. The PCR products (7~10 μL) were resolved on a 1.5~2.0% (*w*/*v*) agarose gel in a 0.5× TBE buffer. A DNA ladder (cat. No. 3428A, lot.AM51847A, Toshima-ku, Tokyo, TaKaRa, Japan) was used to estimate the molecular weight of the bands. The gel was documented using a gel documentation system. The negative control consisted of healthy fish from the control group, whereas the positive control employed commercially available strains, specifically *V. anguillarum* (NCMB 6, ATCC, Changzhou, China) and *S. iniae* (29178, ATCC, China), obtained for scientific purposes.

### 4.2. Bacterial Cultivation and Challenge Test

Before initiating the experimental challenges, bacterial cultures were prepared by growing them on TSA plates (Huankai Microbial -HKM, Co., Ltd., Guangzhou, China, Product Code K097Y, Guangzhou, China) or BHIA (Thermo Fisher Scientific, Product Code BD221841, Waltham, MA, USA), supplemented with 1% NaCl at 28 °C for 24 h. Individual colonies were then incubated overnight in 3 mL of Tryptone Soya Broth (TSB). The following day, 1 mL aliquots of the cultured broth were transferred to fresh TSB to enhance bacterial growth. The colony-forming units (CFU/mL) were determined based on the method described by Miles et al. [131]. To achieve this, bacterial suspensions were subjected to serial 10-fold dilutions in TSB. From each dilution, duplicate 1 mL drops were placed on TSA plates, allowed to air dry, and incubated at 28 °C for 24 h. CFU/mL values were computed using the following formula:$$CFU/mL=\frac{Number\;of\;Colonies\times Dilution\;Factor}{Volume\;Added\;(mL)}$$

The lethal dose 50 (LD50) concentration determined was 1.7 × 10^5^ CFU/mL for *V. anguillarum* and 2 × 10^6^ CFU/mL for *S. iniae*.

A bacterial challenge investigation was performed involving 180 fish, comprising 90 N. Tilapia and 90 E. Sea Bass, across two distinct experiments. Each species was kept in six 70 L tanks, each containing 15 fish exposed to *V. anguillarum* in six different replicates. In a similar fashion, 90 fish were challenged with *S. iniae* in six tanks, with 15 fish per replicate. Control groups for each experiment included 15 fish in three replicates that were not subjected to injections. The fish were administered intraperitoneally with 0.5 mL of bacterial suspension from a 1 mL insulin syringe, containing 1.7 × 10^5^ CFU of *V. anguillarum* per fish and 2 × 10^6^ CFU of *S. iniae* per fish. Control fish were neither challenged nor injected. Observations of the injected fish were made over a 14-day period for signs of infection and rates of mortality. Dead specimens, which included ten fish affected by *V. anguillarum* and nine by *S. iniae*, were collected following the challenge for standard bacteriological analysis.

### 4.3. Experimental Fish for N. Tilapia (O. niloticus) and E. Sea Bass (D. labrax)

#### 4.3.1. Collecting Fish Species

N. Tilapia juveniles (*n* = 500, weight = 125 ± 5.50 g/fish) were obtained from a commercial fish farm, while E. Sea Bass fish (*n* = 550, weight = 115 ± 12.59 g/fish) were obtained from a commercial fish farm (GPS; 31.042120, 30.853177). All fish underwent a comprehensive health screening, including bacteriological and parasitological analyses, and were quarantined with close observation for signs of illness. Additionally, a two-week acclimatization period was implemented to ensure their health and stability prior to the experiments.

#### 4.3.2. Diet Preparation

The fundamental diet, comprising 45% crude protein, was sourced from Aller Aqua, a reputable fish feed manufacturer based in 6 October City, Giza, Egypt and their website at www.aller-aqua.com. The formulation and estimated compositional analysis of this diet were meticulously carried out in accordance with the standards set by AOAC [132]. This adherence ensures the reliability and scientific rigor of the dietary ingredients and nutrient profile utilized in this study.

#### 4.3.3. Acclimation

Before the experiment commenced, the fish were acclimated for two weeks in round fiberglass tanks, each with a capacity of 10 m^3^, which were supplied with dechlorinated tap water and aerators. The tanks were disinfected using a 10% bleach solution followed by thorough rinsing to remove any residues. These fish were acclimatized to laboratory conditions for two weeks at a water temperature of 25 °C, with N. Tilapia in freshwater conditions at 0 ppt salinity and E. Sea Bass in conditions at 30 ppt using added sea salt with dissolved oxygen levels kept above 6 mg/L. Fish density was maintained at approximately 50 kg/m^3^ to ensure adequate space and water quality. Tilapia were kept in 0 ppt salinity to mimic their freshwater habitats, reducing osmotic stress and supporting their immune response, while E. Sea Bass were maintained at 30 ppt to replicate their marine environments, which is essential for their gill function and ion exchange processes. During the acclimatization period, they were fed a commercial diet containing 45% crude protein. At the end of the acclimatization period, the fish in each tank were counted and checked for health, ensuring there were no injuries or deformities. After acclimatization, the fish population was reduced to 445 N. Tilapia and 477 E. Sea Bass.

#### 4.3.4. Fish Groups, Preparation and Sampling

Two separate experiments were conducted for each species, each consisting of three treatment groups, investigated as follows: (1) a control group consisting of 45 fish, divided into three replicates of 15 fish each, housed in fiberglass tanks of 70 L; (2) an infected group with *V. anguillarum*, consisting of 90 fish in six replicates of 15 fish, maintained under the same conditions as the control group except for bacterial infection; (3) an infected group with *S. iniae*, also consisting of 90 fish in six replicates of 15 fish. Blood and tissue samples of five fish (one fish per tank) were collected at the experiment points of 1, 3, and 7 days. The experiment was carefully crafted to achieve highly accurate results, as illustrated in Figure 7.

In this study, fish showing typical hemorrhagic signs were anesthetized and sampled. On days 1, 3, and 7, blood was drawn from the caudal vein of 5~7 fish per group and split into two portions. One portion was mixed with dipotassium EDTA for hematological analysis. The other portion was centrifuged to a separate serum, which was stored at −80 °C for immunological tests.

Additionally, tissue muscle samples of five fish per group were collected at the same intervals and placed in RNAlater for RNA extraction and cDNA synthesis to assess immunity-related genes.

The sampling followed aseptic procedures: fish surfaces were sterilized with 5% Clorox, washed with sterile water, and dissected with a sterilized scalpel. A 1 mg muscle tissue sample was transferred into a sterile Eppendorf tube, labelled, and stored at −20 °C for further analysis.

#### 4.3.5. The Ethical Approval

All aspects of fish handling and sampling in this study received oversight and approval from the Institutional Animal Care and Use Committee at the Faculty of Agriculture, Alexandria University, Egypt, under the approval numbers AU0821045175 and AU082302263130, and all efforts were made to reduce discomfort and distress. The methodologies and experimental protocols adhered rigorously to the guidelines established in the “Guide for the Care and Use of Agricultural Animals in Research and Teaching,” as published by the Federation of Animal Science Societies (FASS, 2010). The full guide is accessible at (https://www.adsa.org/Portals/_default/SiteContent/docs/AgGuide3rd/Ag_Guide_3rd_ed.pdf; accessed on 11 January 2024). Furthermore, this study was conducted in strict accordance with the ARRIVE guidelines, which outline the essential aspects for reporting animal research. These guidelines can be reviewed at arriveguidelines.org.

### 4.4. Biochemical Parameters and Oxidation Assays

#### 4.4.1. Biochemical Parameters

The activity levels of the enzymes, Aspartate Transaminase (AST) and Alanine Transaminase (ALT), in the serum were determined with the help of a BIOLABO Co. kit, Maizy, France, following the protocol established by Henry et al. [133]. To assess renal function markers, such as serum creatinine and urea, a BIOLABO Co. kit was also used. The assessment of serum creatinine was based on the methodology developed by Fabiny and Ertingshausen, [134], while the determination of serum urea followed the procedure described by Tiffany et al. [135].

#### 4.4.2. Oxidation Assays

The levels of Serum Superoxide Dismutase (SOD), Glutathione Peroxidase (GPX), and Catalase (CAT) enzyme activities, along with the concentration of MDA, were measured employing diagnostic reagent kits from Cusabio Biotech Co., Ltd (Houston, TX, USA). The assessments were conducted strictly following the manufacturer’s specified protocols. These procedures allowed for the precise evaluation of antioxidant enzyme activities and lipid peroxidation status in the samples analyzed.

### 4.5. Total and Differential Leukocyte Counts

Total leukocyte counts (WBCs) were determined by manually counting cells using a hemocytometer after dilution with Natt-Herrick’s solution, following the method described by Hrubec et al. [136]. For differential leukocyte counts, blood smears were prepared and stained with Wright’s Giemsa stain, according to Hrubec et al. [136].

### 4.6. Immunological Parameters

Concerning respiratory burst activity, the production of oxygen radicals by blood phagocytes was assessed using the nitroblue tetrazolium (NBT) reduction assay, as outlined by Wijendra and Pathiratne [137]. To perform this assay, 100 μL of whole blood was added to a microtiter plate well along with an equal volume of NBT (1 mg/mL in PBS). The mixture was incubated at room temperature for 30 min. Following incubation, 50 μL of the NBT-blood suspension was mixed with N,N-dimethylformamide in a glass tube and centrifuged, and the supernatant was collected. The optical density (OD) of the supernatant was measured at 540 nm using a spectrophotometer (XYZ-300 Spectrophotometer, Shenzhen Instruments Co., Ltd., Shenzhen, China).

Regarding serum lysozyme activity, it was determined based on the lysis of Micrococcus lysodeikticus, following the method described by Ghareghanipoora et al. [138] with minor adjustments. In brief, 0.25 mL of serum was mixed with 0.75 mL of M. lysodeikticus suspension (0.2 mg/mL in PBS, pH 6.2). This reaction took place at room temperature, and absorbance readings at 450 nm were taken at the start and after 20 min using a photometer. Serum lysozyme concentrations were calculated with reference to a calibration curve prepared using lyophilized chicken egg-white lysozyme.

On the other hand, the levels of immunoglobulin M (IgM) were quantified using a turbidity assay as previously described by Dati and Lammers [139].

For the complement C3 and C4 assay, serum levels of complement C3 and C4 were measured using commercial kits (Elikan, Wenzhou, Zhejiang, China), as referenced by He et al. [140]. The analysis involved examining the increase in turbidity, which corresponds to the immune response involving C3 and C4 after complex formation with antibodies.

### 4.7. Gene Expression

#### 4.7.1. Total Extraction of Total RNA and Synthesis of cDNA

For the extraction of total RNA from fish muscle samples (including the control group and six replicates for each treated group), the protocols established by Chomczynski et al. [141] were utilized. The concentration and purity of the RNA samples were determined using a NanoDrop 2000c spectrophotometer (Thermo, Waltham, Massachusetts, USA). RNA samples were preserved at −80 °C in a Revco horizontal deep freezer (USA) until further processing. cDNA was synthesized by reverse transcription in a 20 μL reaction that contained 3 μL of total RNA, 5 μL of oligo (-dT) primers at 10 pmol/µL, 2.5 μL of dNTP at 10 mM, 2.5 μL of 10x buffer, 0.3 μL M-MULV Reverse Transcriptase (200 units/µL, Biolabs, Hitchin, UK), and 6.7 μL of sterile distilled water to reach a total volume of 20 μL. The reaction mixture was gently mixed by shaking and placed in a thermal cycler (PTC-100TM Programmable Thermal Controller, MJ Research, Inc., Hercules, CA, USA) programmed to run at 37 °C for 90 min, followed by inactivation at 80 °C for 10 min and then cooling at 4 °C. The resulting cDNA-RNA hybrids were stored at −20 °C.

#### 4.7.2. Real-Time Quantitative PCR Approach

In this study, 18 genes, along with the housekeeping gene *β-actin*, as shown in Table 6, were analyzed for their expression in N. Tilapia (*O. niloticus*) and E. Sea Bass (*D. labrax*). These analyses were conducted on both control groups and groups infected with *V. anguillarum* and *S. iniae* on days 1, 3, and 7. The real-time PCR reaction mixture comprised 10 µL of SYBR Green, 1 µL of 10 pm/µL forward primer, 1 µL of 10 pm/µL reverse primer, 1 µL of cDNA (50 ng), and enough sterile dH_2_O to bring the total volume to 20 μL. The PCR reaction conditions included an initial denaturation at 95 °C for 10 min, followed by 45 cycles at 95 °C for 10 s, annealing at 60 °C for 20 s, and elongation at 72 °C for 20 s. Data were captured during the extension phase using the Rotor-Gene 6000 system (Qiagen, Germantown, MD, USA). The difference in quantification was assessed using the ΔΔC_T_ method, where the cycle threshold (CT) values for three replicates were compared between the reference and the experimental samples. The C_T_ values for each gene were normalized against the reference, yielding ΔC_T_ values calculated as follows: ΔC_T_ (_target_) = C_T_ (_target_) − CT (_reference_) and ΔC_T_ (_control_) = C_T_ (_control_) − C_T_ (_reference_). The relative expression of the target gene was determined using the 2^−ΔΔCT^ algorithm, as described by Livak et al. [142], with ΔΔC_T_ calculated as ΔC_T_ (_target_) − ΔCT (control). Threshold cycle values for each gene were established using an automated threshold analysis on the ABI System.

### 4.8. Statistical Analysis

In the process of data analysis, results were expressed as the mean accompanied by the standard error of the mean (mean ± SEM). To evaluate the statistical significance of differences across various groups and control samples, a one-way ANOVA was conducted with a predetermined significance threshold of *p* < 0.05. Subsequently, the Least Significant Difference (LSD) post hoc test was employed to perform pairwise comparisons between specific treatments. To examine gene expression, the data were evaluated using a paired *t*-test and nonparametric methods, utilizing GraphPad Prism version 10.4.0 (GraphPad Software, USA; https://www.graphpad.com/updates, accessed on 11 January 2024). Fold changes were expressed on a log2 scale to emphasize decreased transcription levels. Significance was determined at *p*-values; 0.05 (*), 0.01 (**), and 0.001 (***).

## 5. Conclusions

This study provides compelling insights into the species-specific immune responses of N. Tilapia and E. Sea Bass to infections by *V. anguillarum* and *S. iniae*. The precise molecular identification of these bacterial strains using PCRs underscores the reliability of established detection methodologies and sets the stage for targeted interventions. Biochemically, N. Tilapia exhibited a biphasic immune response, with significant enzyme level changes indicating dynamic physiological adaptations. In contrast, E. Sea Bass demonstrated a delayed yet intense response, particularly at later stages, exemplifying their robust coping mechanisms under stress. Promisingly, oxidative stress assays revealed that E. Sea Bass possesses highly adaptive antioxidant defenses, reflected in their dynamic enzyme activity changes that suggest promising resilience against bacterial stressors. In both species, elevated levels of cytokines, such as *IL-6* and chemokine *CXCL-10*, highlight their potent roles in sustaining immune engagement, offering potential targets for enhancing disease resistance. The gene expression analyses provided further promising data, with significant upregulation of critical immune molecules like *TNF-α* and *IFN-γ* underscoring their importance in pro-inflammatory pathways. Additionally, the modulation of AMPs and Galectin genes reveals species-specific regulatory pathways that could be harnessed to optimize immune responses. Overall, these findings are promising for advancing aquaculture health strategies. By tailoring management practices to the unique immune capabilities of each species, there is potential to enhance resilience against pathogenic challenges, improving overall fish health and productivity in aquaculture systems.

## Figures and Tables

**Figure 1 ijms-25-12829-f001:**
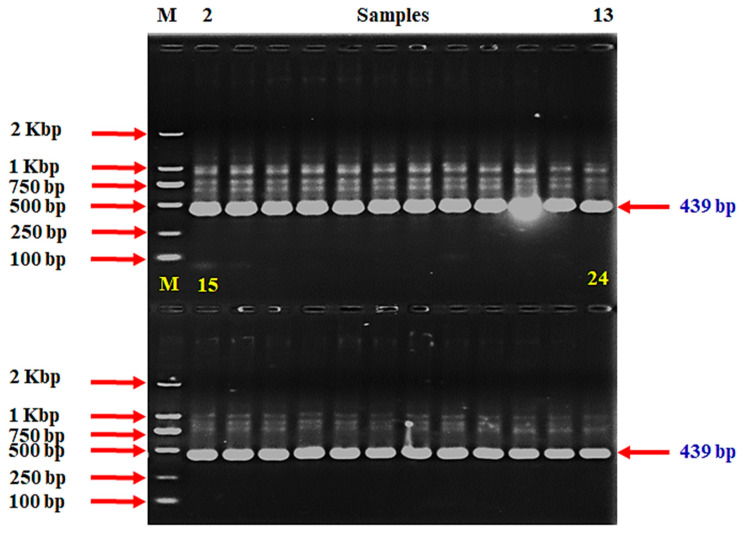
PCR amplification results for *empA* gene fragment (439 bp) from *V. anguillarum* isolated from infected fish samples. Columns represent M for the 2 kbp DNA marker, and Columns 3–12 and 16–23 for the infected samples, and Columns 2, 13, 15 and 24 for positive control.

**Figure 2 ijms-25-12829-f002:**
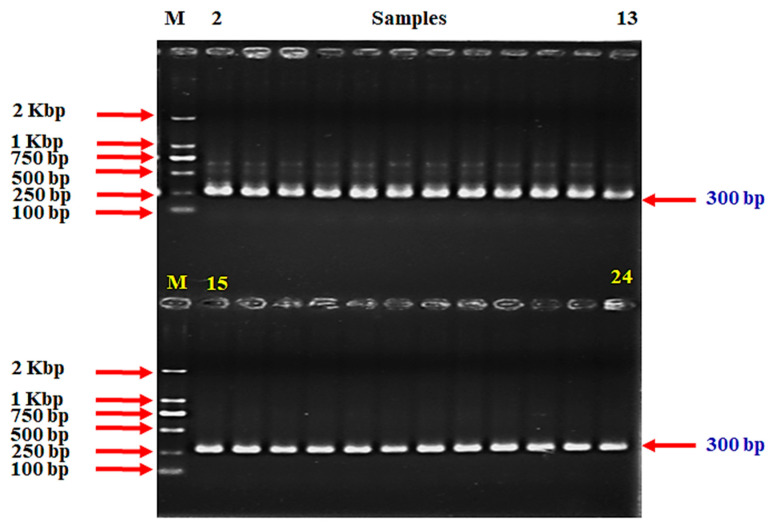
PCR amplification results for *Sin* gene fragment (300 bp) from *S. iniae* isolated from infected fish samples. Columns represent M for the 2 kbp DNA marker, and Columns 3–12 and 16–23 for the infected samples, and Columns 2, 13, 15 and 24 for positive control.

**Figure 3 ijms-25-12829-f003:**
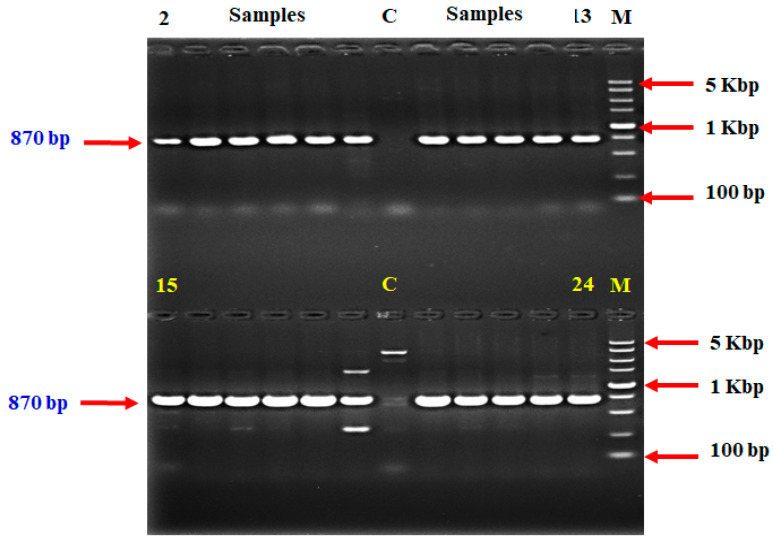
PCR amplification results for *lctO* gene fragment (870 bp) from *S. iniae* isolated from infected fish samples. Columns represent M for the 5 kbp DNA marker, and columns 3–12 and 16–23 for the infected samples C column; negative control, and Columns 2, 13, 15 and 24 for positive control.

**Figure 4 ijms-25-12829-f004:**
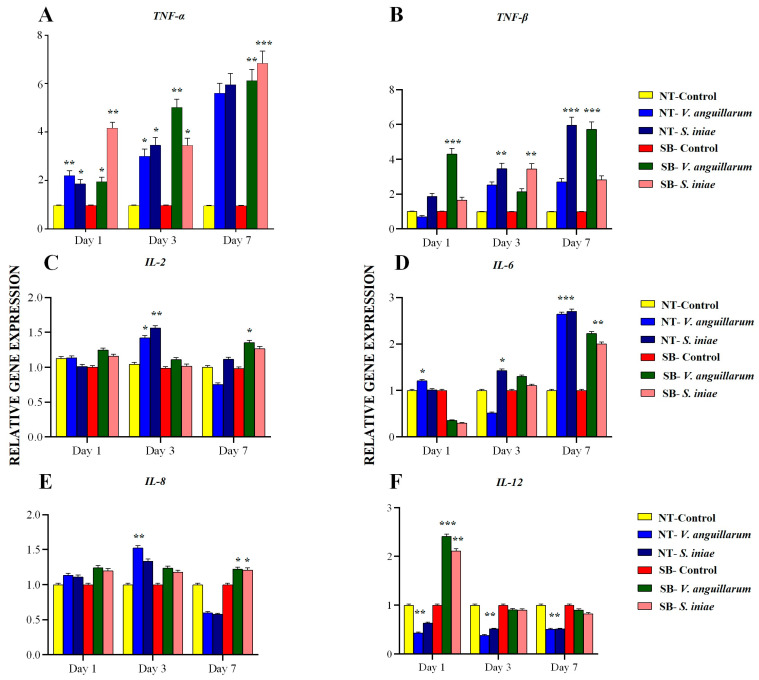
The relative expression levels of *TNF-α* (**A**), *TNF-β* (**B**), *IL-2* (**C**), *IL-6* (**D**), *IL-8* (**E**), and *IL-12* (**F**) genes in muscle tissues of *O. niloticus* and *D. labrax*. These fish were exposed to *V. anguillarum* and *S. iniae*, with analyses conducted on days 1, 3, and 7 following the challenge. The data were normalized using *β-actin* as the reference gene. Statistical analysis was performed using a paired t-test, and results are shown as fold changes relative to the control group. The values are presented as the mean ± SEM, with sample sizes of *n* = 5~7 for each group. Asterisks indicate significant differences between groups: * *p* < 0.05, ** *p* < 0.01, and *** *p* < 0.001.

**Figure 5 ijms-25-12829-f005:**
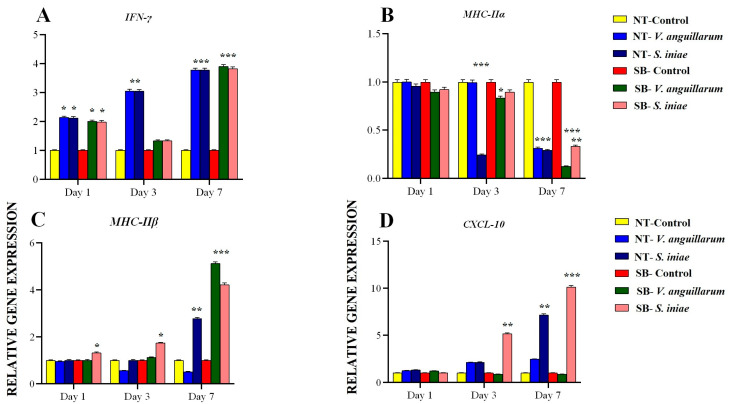
The relative expression levels of *IFN-γ* (**A**), *MHC-IIα* (**B**), *MHC-IIβ* (**C**), *CXCL-10* (**D**), *CD4-12* (**E**), and *Pleurocidin* (**F**) genes in muscle tissues of *O. niloticus* and *D. labrax*. These fish were exposed to *V. anguillarum* and *S. iniae*, with analyses conducted on days 1, 3, and 7 following the challenge. The data were normalized using *β-actin* as the reference gene. Statistical analysis was performed using a paired *t*-test, and results are shown as fold changes relative to the control group. The values are presented as the mean ± SEM, with sample sizes of *n* = 5~7 for each group. Asterisks indicate significant differences between groups: * *p* < 0.05, ** *p* < 0.01, and *** *p* < 0.001.

**Figure 6 ijms-25-12829-f006:**
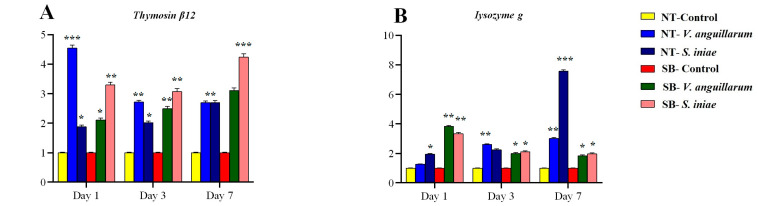
The relative expression levels of *Thymosin β12* (**A**), *Iysozyme g* (**B**), *Leap 2* (**C**), *β-defensin* (**D**), *Galectin-8* (**E**), and *Galectin-9* (**F**) genes in muscle tissues of *O. niloticus* and *D. labrax*. These fish were exposed to *V. anguillarum* and *S. iniae*, with analyses conducted on days 1, 3, and 7 following the challenge. The data were normalized using *β-actin* as the reference gene. Statistical analysis was performed using a paired *t*-test, and results are shown as fold changes relative to the control group. The values are presented as the mean ± SEM, with sample sizes of *n* = 5~7 for each group. Asterisks indicate significant differences between groups: * *p* < 0.05, ** *p* < 0.01, and *** *p* < 0.001.

**Figure 7 ijms-25-12829-f007:**
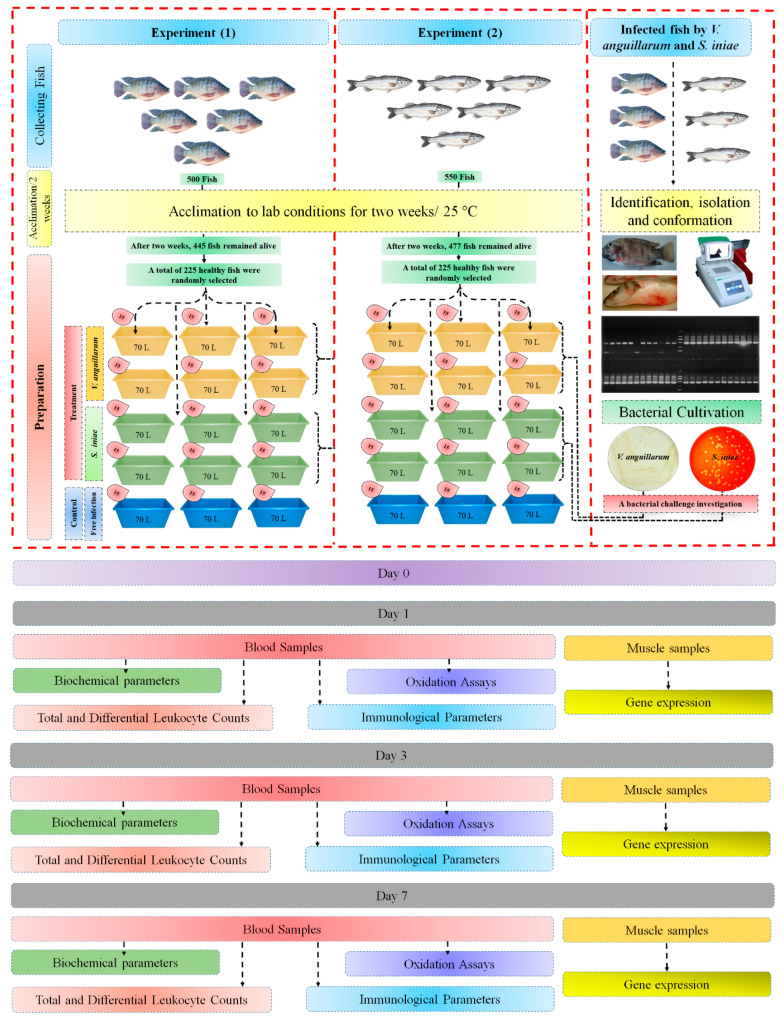
Experimental Design Overview. Schematic representation of the experimental setup for N. Tilapia (*O. niloticus*) and E. Sea Bass (*D. labrax*) to study physiological and molecular responses. Fish were sourced, screened for health, and acclimated for two weeks under species-specific salinity conditions (0 ppt for N. Tilapia, 30 ppt for E. Sea Bass). The study comprised three groups: a control group, and two groups infected with *V. anguillarum* and *S. iniae*. Each group contained a set number of replicates, with fish sampled on days 1, 3, and 7 to analyze response dynamics.

**Table 1 ijms-25-12829-t001:** Biochemical indices of N. Tilapia and E. Sea Bass infected by *V. anguillarum* and *S. iniae*.

Groups	ALT (U/L)	AST (U/L)	Urea (mg/dL)	Creatinine (mg/dL)
N. Tilapia				
Control group day 1	12.33 ± 1.52 ^a^	33.12 ± 3.00 ^a^	15.10 ± 3.00 ^a^	0.45 ± 0.23 ^a^
Infected group day 1/*V. anguillarum*	5.89 ± 1.32 ^b^	19.20 ± 4.10 ^b^	16.86 ± 1.00 ^a^	0.34 ± 0.10 ^b^
Infected group day 1/*S. iniae*	6.00 ± 1.20 ^b^	22.00 ± 3.20 ^b^	16.50 ± 1.90 ^a^	0.33 ± 0.11 ^b^
Control group day 3	12.40 ± 1.50 ^a^	34.00 ± 3.05 ^a^	15.50 ± 2.90 ^a^	0.46 ± 0.22 ^a^
Infected group day 3/*V. anguillarum*	5.17 ± 0.76 ^b^	10.72 ± 1.51 ^c^	19.33 ± 2.5 ^b^	0.32 ± 0.087 ^b^
Infected group day 3/*S. iniae*	5.50 ± 1.00 ^b^	25.00 ± 3.40 ^b^	17.00 ± 2.00 ^a^	0.35 ± 0.10 ^b^
Control group day 7	13.00 ± 1.40 ^a^	35.00 ± 3.20 ^a^	16.00 ± 3.00 ^a^	0.44 ± 0.21 ^a^
Infected group day 7/*V. anguillarum*	13.10 ± 1.05 ^a^	42.90 ± 3.11 ^d^	17.23 ± 1.23 ^ad^	0.37 ± 0.056 ^ab^
Infected group day 7/*S. iniae*	14.00 ± 1.00 ^a^	45.00 ± 3.00 ^d^	18.00 ± 2.50 ^b^	0.38 ± 0.12 ^ab^
E. Sea Bass				
Control group day 1	11.45 ± 2.05 ^a^	20.21 ± 2.00 ^a^	12.52 ± 1.10 ^a^	0.29 ± 0.08 ^a^
Infected group day 1/*V. anguillarum*	11.67 ± 1.51 ^a^	20.12 ± 1.00 ^a^	13.67 ± 1.09 ^a^	0.29 ± 0.09 ^a^
Infected group day 1/*S. iniae*	11.50 ± 1.60 ^a^	21.00 ± 2.10 ^a^	13.00 ± 1.15 ^a^	0.30 ± 0.09 ^a^
Control group day 3	11.20 ± 1.90 ^a^	21.00 ± 2.50 ^a^	12.70 ± 1.20 ^a^	0.29 ± 0.09 ^a^
Infected group day 3/*V. anguillarum*	7.67 ± 0.58 ^b^	20.10 ± 1.50 ^a^	13.53 ± 1.28 ^a^	0.48 ± 0.17 ^b^
Infected group day 3/*S. iniae*	9.00 ± 1.00 ^ab^	22.50 ± 3.00 ^a^	14.00 ± 1.40 ^a^	0.32 ± 0.11 ^a^
Control group day 7	11.50 ± 1.90 ^a^	22.00 ± 2.50 ^a^	13.00 ± 1.20 ^a^	0.30 ± 0.09 ^a^
Infected group day 7/*V. anguillarum*	32.00 ± 3.20 ^c^	44.00 ± 3.10 ^c^	21.00 ± 2.70 ^c^	0.60 ± 0.25 ^c^
Infected group day 7/*S. iniae*	33.5 ± 3.20 ^c^	43.13 ± 3.05 ^c^	20.54 ± 2.67 ^c^	0.55 ± 0.21 ^c^

Data are expressed as mean ± SEM (*n* = 5~7/each replicate). ^abcd^ Letters indicate statistical significance: groups sharing the same letter are not significantly different (*p* > 0.05). *p* < 0.05 indicates significant differences among groups. ALT: Alanine Transaminase; AST: Aspartate Transaminase.

**Table 2 ijms-25-12829-t002:** Enzymatic antioxidant activities of N. Tilapia and E. Sea Bass infected by *V. anguillarum* and *S. iniae*.

Groups	MDA (IU/I)	SOD (IU/I)	GPX (IU/I)	CAT (IU/I)
N. Tilapia				
Control group day 1	9.22 ± 0.68 ^a^	1.24 ± 1.10 ^a^	2.58 ± 0.19 ^a^	567 ± 8.00 ^a^
Infected group day 1/*V. anguillarum*	8.11 ± 0.12 ^b^	1.62 ± 0.14 ^a^	3.67 ± 0.32 ^a^	584.70 ± 19.20 ^a^
Infected group day 1/*S. iniae*	8.55 ± 0.20 ^ab^	1.45 ± 0.10 ^a^	3.50 ± 0.25 ^a^	580 ± 15.00 ^a^
Control group day 3	9.30 ± 0.70 ^a^	1.30 ± 0.12 ^a^	2.60 ± 0.18 ^a^	570.56 ± 10.00 ^a^
Infected group day 3/*V. anguillarum*	5.11 ± 1.20 ^bc^	1.91 ± 0.14 ^b^	5.02 ± 0.19 ^bc^	599.30 ± 13.01 ^b^
Infected group day 3/*S. iniae*	6.00 ± 0.50 ^c^	2.00 ± 0.20 ^b^	4.80 ± 0.23 ^b^	590.34 ± 12.00 ^b^
Control group day 7	11.01 ± 0.72 ^acd^	1.42 ± 0.34a ^c^	1.97 ± 1.20 ^bcd^	429.33 ± 14.12 ^bcd^
Infected group day 7/*V. anguillarum*	10.50 ± 0.60 ^d^	1.50 ± 0.12 ^c^	2.00 ± 0.15 ^c^	430.50 ± 12.00 ^c^
Infected group day 7/*S. iniae*	12.00 ± 0.50 ^d^	1.55 ± 0.15 ^c^	2.10 ± 0.18 ^c^	435.43 ± 13.00 ^c^
E. Sea Bass				
Control group day 1	9.23 ± 0.88 ^a^	1.24 ± 1.00 ^a^	2.70 ± 0.15 ^a^	591 ± 8.00 ^a^
Infected group day 1/*V. anguillarum*	5.80 ± 0.5 ^b^	2.35 ± 0.24 ^b^	3.76 ± 0.051 ^b^	646 ± 8.56 ^b^
Infected group day 1/*S. iniae*	6.00 ± 0.60 ^b^	2.25 ± 0.20 ^b^	3.70 ± 0.20 ^b^	640 ± 9.00 ^b^
Control group day 3	9.50 ± 0.80 ^a^	1.30 ± 0.15 ^a^	2.80 ± 0.18 ^a^	600 ± 8.00 ^a^
Infected group day 3/*V. anguillarum*	7.11 ±0.75 ^b^	1.93 ± 0.26 ^bc^	2.88 ± 0.29 ^ac^	585 ± 22.25 ^ac^
Infected group day 3/*S. iniae*	7.50 ± 0.70 ^b^	2.10 ± 0.22 ^bc^	3.00 ± 0.25 ^b^	590 ± 15.00 ^bc^
Control group day 7	10.00 ± 0.50 ^a^	1.40 ± 0.10 ^a^	2.90 ± 0.12 ^a^	580 ± 9.00 ^a^
Infected group day 7/*V. anguillarum*	11.82 ± 0.30 ^bcd^	1.44 ± 0.51 ^acd^	2.57 ± 0.34 ^ad^	566 ± 9.23 ^bcd^
Infected group day 7/*S. iniae*	13.00 ± 0.40 ^cd^	1.50 ± 0.30 ^cd^	2.70 ± 0.22 ^ad^	570 ± 10.00 ^cd^

Data are expressed as Mean ± SEM (*n* = 5~7/each replicate). ^abcd^ Letters indicate statistical significance: groups sharing the same letter are not significantly different (*p* > 0.05). *p* < 0.05 indicates significant differences among groups. MDA; Malondialdehyde, SOD; Activities of Serum Superoxide Dismutase, GPX; Glutathione peroxidase, CAT; Catalase.

**Table 3 ijms-25-12829-t003:** Leukocytic counts in N. Tilapia and E. Sea Bass.

Groups	Total Leukocytic Counts (10^3^/μL)	Lymphocyte (10^3^/μL)	Neutrophil (10^3^/μL)	Monocyte (10^3^/μL)
N. Tilapia				
Control group day 1	22.69 ± 3.01 ^b^	13.34 ± 1.12 ^b^	8.27 ± 1.19 ^b^	1.71 ± 0.26 ^b^
Infected group day 1/*V. anguillarum*	35.45 ± 2.78 ^a^	17.21 ± 0.85 ^a^	13.30 ± 1.39 ^a^	3.95 ± 1.10 ^a^
Infected group day 1/*S. iniae*	31.40 ± 3.17 ^a^	19.50 ± 0.63 ^a^	12.95 ± 1.20 ^a^	2.28 ± 0.39 ^b^
Control group day 3	22.64 ± 3.17 ^b^	13.40 ± 1.18 ^b^	8.27 ± 1.28 ^b^	1.82 ± 0.36 ^b^
Infected group day 3/*V. anguillarum*	21.70 ± 2.35 ^c^	11.10 ± 1.45 ^c^	8.10 ± 0.20 ^b^	1.05 ± 0.11 ^b^
Infected group day 3/*S. iniae*	29.15 ± 2.80 ^b^	14.50 ± 2.00 ^b^	10.80 ± 2.05 ^a^	1.49 ± 0.49 ^b^
Control group day 7	23.10 ± 3.25 ^b^	13.34 ± 1.08 ^b^	8.27 ± 1.28 ^b^	1.71 ± 0.26 ^b^
Infected group day 7/*V. anguillarum*	35.92 ± 3.35 ^a^	15.05 ± 3.00 ^a^	17.50 ± 2.40 ^a^	5.30 ± 0.25 ^a^
Infected group day 7/*S. iniae*	35.20 ± 2.22 ^a^	13.40 ± 2.20 ^b^	18.50 ± 3.10 ^a^	3.70 ± 0.12 ^a^
E. Sea Bass				
Control group day 1	25.10 ± 2.60 ^b^	14.50 ± 1.20 ^b^	9.10 ± 1.30 ^b^	1.50 ± 0.20 ^b^
Infected group day 1/*V. anguillarum*	35.20 ± 3.00 ^a^	18.80 ± 1.00 ^a^	14.40 ± 1.60 ^a^	4.80 ± 1.35 ^a^
Infected group day 1/*S. iniae*	33.00 ± 3.20 ^a^	17.50 ± 0.45 ^a^	13.50 ± 1.50 ^a^	2.60 ± 0.45 ^b^
Control group day 3	25.10 ± 2.60 ^b^	14.50 ± 1.20 ^b^	9.10 ± 1.30 ^b^	1.50 ± 0.20 ^b^
Infected group day 3/*V. anguillarum*	21.00 ± 2.40 ^c^	12.00 ± 1.50 ^c^	9.50 ± 0.25 ^b^	1.20 ± 0.15 ^b^
Infected group day 3/*S. iniae*	27.60 ± 3.20 ^b^	15.90 ± 1.80 ^a^	12.50 ± 2.20 ^a^	1.70 ± 0.55 ^b^
Control group day 7	25.10 ± 2.60 ^b^	14.50 ± 1.20 ^b^	9.10 ± 1.30 ^b^	1.50 ± 0.20 ^b^
Infected group day 7/*V. anguillarum*	38.00 ± 3.60 ^a^	16.60 ± 3.10 ^a^	17.10 ± 2.60 ^a^	6.10 ± 0.28 ^a^
Infected group day 7/*S. iniae*	35.80 ± 2.30 ^a^	14.00 ± 2.40 ^b^	19.30 ± 3.20 ^a^	3.80 ± 0.15 ^a^

Data are expressed as mean ± SEM (*n* = 5~7/each replicate). ^abc^ Letters indicate statistical significance: groups sharing the same letter are not significantly different (*p* > 0.05). *p*-value < 0.05 indicates significant differences among groups. These values represent a hypothetical immune response profile for different infections and conditions in both fish species.

**Table 4 ijms-25-12829-t004:** Comparative analysis of immune response parameters in N. Tilapia and E. Sea Bass following bacterial infections.

Groups	Respiratory Burst Activity (O.D. at 630 nm)	Serum Lysozyme Activity (μg/mL)	Immunoglobulin M (mg/dL)	Complement C3 (mg/dL)	Complement C4 (mg/dL)
N. Tilapia					
Control group day 1	0.20 ± 0.02 ^b^	8.00 ± 2.50 ^b^	75.00 ± 5.00 ^b^	4.50 ± 0.50 ^b^	0.90 ± 0.10 ^b^
Infected group day 1/*V. anguillarum*	0.28 ± 0.03 ^a^	11.50 ± 2.80 ^a^	90.00 ± 6.00 ^a^	8.00 ± 1.00 ^a^	1.60 ± 0.20 ^a^
Infected group day 1/*S. iniae*	0.25 ± 0.02 ^a^	10.50 ± 2.00 ^a^	85.00 ± 4.50 ^a^	6.50 ± 0.80 ^a^	1.20 ± 0.15 ^a^
Control group day 3	0.18 ± 0.02 ^b^	8.00 ± 2.50 ^b^	73.00 ± 4.20 ^b^	4.50 ± 0.50 ^b^	0.90 ± 0.10 ^b^
Infected group day 3/*V. anguillarum*	0.33 ± 0.04 ^a^	13.00 ± 3.00 ^a^	88.00 ± 3.70 ^a^	9.50 ± 1.20 ^a^	3.80 ± 0.40 ^a^
Infected group day 3/*S. iniae*	0.30 ± 0.03 ^a^	12.00 ± 2.50 ^a^	83.00 ± 5.10 ^b^	7.00 ± 0.90 ^a^	2.10 ± 0.20 ^a^
Control group day 7	0.20 ± 0.02 ^b^	8.00 ± 2.50 ^b^	70.00 ± 3.50 ^c^	4.50 ± 0.50 ^b^	0.90 ± 0.10 ^b^
Infected group day 7/*V. anguillarum*	0.40 ± 0.05 ^a^	10.50 ± 3.20 ^a^	85.00 ± 5.50 ^a^	3.00 ± 0.50 ^a^	1.80 ± 0.20 ^a^
Infected group day 7/*S. iniae*	0.38 ± 0.04 ^a^	11.00 ± 2.70 ^a^	80.00 ± 3.80 ^b^	4.00 ± 0.60 ^a^	1.50 ± 0.15 ^a^
E. Sea Bass					
Control group day 1	0.22 ± 0.02 ^b^	9.00 ± 3.00 ^b^	78.00 ± 4.70 ^b^	4.80 ± 0.60 ^b^	0.95 ± 0.10 ^b^
Infected group day 1/*V. anguillarum*	0.30 ± 0.04 ^a^	12.00 ± 3.50 ^a^	92.00 ± 6.50 ^a^	8.50 ± 1.10 ^a^	1.70 ± 0.25 ^a^
Infected group day 1/*S. iniae*	0.27 ± 0.03 ^a^	11.00 ± 2.80 ^a^	87.00 ± 5.20 ^a^	6.80 ± 0.90 ^a^	1.30 ± 0.20 ^a^
Control group day 3	0.21 ± 0.02 ^b^	9.00 ± 3.00 ^b^	76.00 ± 5.00 ^b^	4.80 ± 0.60 ^b^	0.95 ± 0.10 ^b^
Infected group day 3/*V. anguillarum*	0.35 ± 0.05 ^a^	13.50 ± 3.20 ^a^	90.00 ± 4.80 ^a^	9.00 ± 1.30 ^a^	4.00 ± 0.50 ^a^
Infected group day 3/*S. iniae*	0.32 ± 0.04 ^a^	12.50 ± 3.00 ^a^	84.00 ± 4.90 ^b^	7.20 ± 1.00 ^a^	2.30 ± 0.30 ^a^
Control group day 7	0.22 ± 0.02 ^b^	9.00 ± 3.00 ^b^	74.00 ± 4.20 ^b^	4.80 ± 0.60 ^b^	0.95 ± 0.10 ^b^
Infected group day 7/*V. anguillarum*	0.42 ± 0.05 ^a^	11.00 ± 3.50 ^a^	87.00 ± 5.80 ^a^	3.20 ± 0.70 ^a^	1.90 ± 0.25 ^a^
Infected group day 7/*S. iniae*	0.40 ± 0.04 ^a^	11.50 ± 3.20 ^a^	82.00 ± 4.10 ^b^	4.20 ± 0.70 ^a^	1.60 ± 0.20 ^a^

Data are expressed as mean ± SEM (*n* = 5~7/each replicate). ^abc^ Letters indicate statistical significance: groups sharing the same letter are not significantly different (*p* > 0.05). *p* < 0.05 indicates significant differences among groups. These values represent a hypothetical immune response profile for different infections and conditions in both fish species.

**Table 5 ijms-25-12829-t005:** Oligonucleotide primers utilized in this study to confirm infections caused by *V. anguillarum* and *S. iniae*.

Target Gene	Species	Primers 5′→3′	Fragment Size	Ref.
*empA*	*V. anguillarum*	F: 5′-CAGGCTCGCAGTATTGTGC-3′	439 bp	[47]
		R: 5′-CGTCACCAGAATTCGCATC-3′		
*Sin*	*S. iniae*	F: 5′-CTAGAGTACACATGTAGCTAAG-3′	300 bp	[48]
		R: 5′-GGATTTTCCACTCCCATTAC-3′		
*lctO*		F: 5′-AAGGGGAAATCGCAAGTGCC-3′	870 bp	[49]
		R: 5′-ATATCTGATTGGGCCGTCTAA-3′		

**Table 6 ijms-25-12829-t006:** The specific primers of defense genes used in real-time PCR.

No.	Target Gene	Primers 5′→3′	Ref.
**Standard**	*β-actin* (housekeeping)	F: 5′-ATGCCATTCTCCGTCTTGACTTG-3′	[143]
		R: 5′-GAACCTAAGCCACGATACCA-3′	
**1**	*TNF-α*	F: 5′-CCACACCACGTTGAGGCAGATCA-3′	[144]
		R: 5′-CCTTGACCGCTTCTCCACTCCA-3′	
**2**	*TNF-β*	F: 5′- GGTGCCCAGAGATGGCTTGTA-3′	[28]
		R: 5′-TTGTGTGGATTGATGAGAGGAGAGT-3′	
**3**	*IL-2*	F: 5′-AAGAGTCATCAGAAGAGGAAA-3′	[143]
		R: 5′-AACCTT GGGCATGTAGAAGT-3′	
**4**	*IL-6*	F: 5′-CCAGGATCCCAGCTATGAACTCCCTCTTC-3′	[145]
		R: 5′-GGAGAATTCGCTACTTCATCCGAATGACTC-3′	
**5**	*IL-8*	R: 5′-CGGAATTCATGAAGGCTGCAACT-3′	[113]
		R: 5′-CCCTCGAGTCAGTTTTGCTGTTTG-3′	
**6**	*IL-12*	F: 5′-CACCACCTGCCCCACCTCAG-3′	[146]
		R: 5′-CTACGAAGAACTCAGATAG-3′	
**7**	*IFN-γ-*	F: 5′-TGCACGAAGTGAAAGACCAAA-3′	
		R: 5′-TTAAGGTCCAGCAGCTCAGTGA-3′	
**8**	*MHC-IIα*	F: 5′-GGACAGGTTTGAAGCCAGAGTT-3′	[46]
		R: 5′-CGGGAAGGAGATTAAAGGAGGT-3′	
**9**	*MHC-IIβ*	F: 5′-CGGGAAGGAGATTAAAGGAGGT-3′	[113]
		R: 5′-GTTTGGTGAAGCTGGCGTGT-3′	
**10**	*CXCL-10*	F: 5′-ACAGGCCAGGACCAGTGTAAGG-3′	[113]
		R: 5′-CAAGTTGCACTCGCAGGATGAA-3′	
**11**	*CD4-L2*	F: 5′-GCAGGGCACGGATAGATGGA-3′	[113]
		R: 5′-TGGGTTCGCAGAGGCTGATAC-3′	
**12**	*Pleurocidin*	F: 5′-GATGAAGTGTATCGTGGTG-3′	[147]
		R: 5′-TAGGCTGTCCTGGGTT-3′	
**13**	*Thymosin β12*	F: 5′-CGACATTTCAGAAGTGACCAG-3′	
		R: 5′-CTCTTTTGTAGGCAGGGGATT-3′	
**14**	*Iysozyme g*	F: 5′-CATGGCAAAGACTGATGCGG-3′	
		R: 5′-TGTTCCTCACTGTCCCATGC-3′	
**15**	*Leap 2*	F: 5′-CAAAGGAAAGCAGCAGTAGCA-3′	
		R: 5′-CATAGTTGTTCTGGCAGTAAGC-3′	
**16**	*β-defensin*	F: 5′-TGTGCTTCTCCTGATGCTCG-3′	
		R: 5′-TGTGACATCTTCCAGGCGTC-3′	
**17**	*Galectin-8*	F: 5′-GGCGACTTGAGTGTTCC-3′	
		R: 5′-TTTCTTCAGACGAGGGTT-3′	
**18**	*Galectin-9*	F: 5′-ATTCCCTGCTGGCTCTAC-3′	
		R: 5′-TCTACTTTCCCGCCTACTG-3′	

## Data Availability

All data generated or analyzed during this study are included in this manuscript and Appendix A.

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
