# Peer review of "Expression and Immune Response Profiles in Nile Tilapia (Oreochromis niloticus) and European Sea Bass (Dicentrarchus labrax) During Pathogen Challenge and Infection"

_ijms, 2024, doi:10.3390/ijms252312829_

Round 1

Reviewer 1 Report

Comments and Suggestions for Authors

This study investigated the effects of two common bacterial pathogens on Nile Tilapia and European Sea Bass through biochemical and immunological parameters. The research methodology adopted in this study is sound, ensuring the reliability and validity of the obtained results. And the discussion was comprehensive, providing in-depth insight. This research expands the understanding of the response mechanisms of N. Tilapia and E. Sea Bass to bacterial infections. However, some issues still need to be concerned.

1. There are many Latin names of species in the full text that are not in italics. Please check.

2. Line 709: Please further explain why the results of this study are contrary to those of Rahimikia's. In my opinion, they have similar results, but the conclusions drawn are different.

3. Line 733: It is mentioned here that bacterial infections may cause kidney damage. Are there any relevant studies on tilapia or sea bass? If so, please add them.

4. Line 740: Are there any relevant studies that have proved this speculation? If there are, please add them.

5. Line 758: This section lacks the support of research on other fish species. Please add relevant content.

6. There are many repetitions in sections 4.4 and 4.5 compared with the Results section. Please reduce them as appropriate.

7. The discussion contents in sections 4.6.2, 4.6.3 and 4.6.4 are a little bit redundant. Please keep the contents that are more relevant to this study. In addition, muscle tissues were sampled in this study, while most of the cited references are about immune-related visceral organs. It is necessary to add relevant studies on samples of muscle tissues in this part.

8. Line 949: The content citing Elbahnaswy 's research results is too detailed. Please only keep the main conclusions relevant to this study.

Author Response

Dear Reviewer (1), We sincerely thank you for the valuable feedback and constructive insights. Your comments have been instrumental in enhancing the clarity, rigor, and overall quality of our manuscript. We appreciate your time and expertise in helping us improve our work. Please see the attachment (Point-by-Point response to the two reviewers and editors). 

Notes;

  1. We have provided two copies of the manuscript:
  2. 1.1.MS_Clean version (with the main submission)
  3. 1.2.MS_Tracking version (as Related files/PDF).
  4. The changes suggested by Reviewer #1 are highlighted in yellow, while those suggested by Reviewer #2 are highlighted in green, and those suggested by the Editor are highlighted in blue.
  5. Additionally, we have uploaded a file containing our responses to the Editor and Reviewers, addressing each point individually.

Sincerely yours,

Reviewer 2 Report

Comments and Suggestions for Authors

The manuscript entitled "Expression and Immune Response Profiles in Nile Tilapia (O.niloticus) and European Sea Bass (D. labrax) During Pathogen Challenge and Infection", by Saleh et al. aims to identify signatures of impact of infection on gene expression, immunological profiles, biochemical indices, and antioxidant status in two highly commercial aquaculture species. It is an interesting study, identifying potential molecular mechanisms that are related to disease vulnerability in aquaculture.

As a general comment, the ms is too long, including Abstract. If you don't have any special argument on that, I'd suggest reducing the ms, keeping only the important things for the study. Also, as I say at the end, I'd suggest the author discuss aquaculture practices that also enhance the presence of infectious diseases in conjunction with abiotic factors, such as temperature.

I pointed out a couple of things related to acclimation and molecular techniques. Overall, I'd suggest the authors to consider my comments that have only the purpose to improve the study.

Introduction

line 73-75: Please provide a more recent yearly value, and references. Also, please provide the relevant value of production for Tilapia as well.

line 96-98: 21 reference is the same as 22. Also, 21/22 and 23 references refer to streptococcosis, but Vibrio causes vibriosis. So, please use relevant references for V.anguillarum.

line 129: Reference 32 is referred to a different species (flounder). What is the correlation with Tilapia or sea bass?

line 130-140: This paragraph refers to mandarin fish and to zebrafish. There are plenty of studies regarding AMPs in cultured species, such as S.aurata. I'd suggest the authors reconsider this paragraph (and probably elsewhere) and refer to aquaculture species for consumption and only.

lines 146-155: Are there any other studies that deal with the same question (gene expression and immune response)? The bibliography is tremendously rich in such studies. Please provide a few examples/studies, and according to their findings build up your hypotheses. Right now the hypotheses/aim section is vague without any specific questions that need to be addressed.

M&Ms

lines 158-161: When was the season of the strains collection? How did you obtain the sample? Please be specific.

lines 166-168: What was the positive control? Did you have strains from other experiments, or from any kit?

lines 169-176: Please provide PCR conditions, reagents etc.

lines 189-190: Why the concentration is different? Please explain.

lines 191-202 and lines 228-241: Are these two different experiments, or the same? Please provide it only once if it's the same. If it's not the same, what is the purpose of the 191-202 description?

lines 208-210: Please provide evidence and tests that you performed to identify the health status of the collected fish samples. How were you sure that these specimens were not prior infected with V.anguillarum and S.iniae, or any other bacterial/parasite/virus?

lines 219-226: Please describe what kind of disinfection you did on these tanks. Also, what water did you use (sea water or tap-water and you added salt)? Please provide temperature, salinity, oxygen and density values for the acclimation stage. Did you use high temperatures, likewise in the summer? Normal ones?

lines 321-337: Did you check for the sensitivity and reproducibility of the qPCR method? [e.g. using five 10-fold serial dilutions of each DNA template (10 ng/μL to 1 pg/μL)]. Not quite sure as it's written, but did you also use technical replicates apart from the biological replicates?

Results

lines 355-364: You should also include negative controls of the PCR-reaction, and also negative biological samples controls. Doing that, you are sure that the amplification is not an artifact, or a contamination. Please provide a gel-photo with the previous negative controls. Also, did you check a couple of the 3-12/16-23 samples through Sanger sequence for the accuracy of the results?

Figure 4: What's that band at the negative Control, 2nd row? What kind of negative is that? PCR-negative, or bacteria-free specimen?

lines 377-610: Great analytical presentation of the results. Just one question. Did you check for variation between the different tanks? Please provide a relevant test, that there wasn't any significant variation between the biochemical/immunological/gene expression profiles values among the different control and test tanks for both species and each treatment.

Discussion

This section is rather big (almost 9 pages!). I'd suggest authors to reduce it and stick to the important associations between the analysed profiles and the immune response. Authors propose a molecular mechanism of immune response for these two specific bacterias -which is great - but as far as I understand the fishes will get infected and die no matter what? Are there any other potential solutions to avoid such bacterias? I'd suggest authors discuss and argue more on aquaculture techniques and methods that might increase the outburst of such bacterias. Also, in the discussion they do not correlate temperature with different profiles.

Author Response

Dear Reviewer (2), We are deeply grateful for your insightful feedback and constructive suggestions. Your thorough review has significantly contributed to improving the clarity and quality of our manuscript. We greatly value your expertise and the time you dedicated to helping us enhance our work. Thank you sincerely for your support. Please see the attachment (Point-by-Point response to the two reviewers and editors). 

Notes;

  1. We have provided two copies of the manuscript;
  2. 1.1.MS_Clean version (with the main submission)
  3. 1.2.MS_Tracking version (as Related files/PDF).
  4. The changes suggested by Reviewer #1 are highlighted in yellow, while those suggested by Reviewer #2 are highlighted in green, and those suggested by the Editor are highlighted in blue.
  5. Additionally, we have uploaded a file containing our responses to the Editor and Reviewers, addressing each point individually.

    Sincerely yours,
